# On Learning Ising Models under Huber's Contamination Model

**Adarsh Prasad**[*][†] **Vishwak Srinivasan**[*][†] **Sivaraman Balakrishnan**[‡][†] **Pradeep Ravikumar**[†]

adarshp@andrew.cmu.edu, vishwaks@cs.cmu.edu
siva@stat.cmu.edu, pradeepr@cs.cmu.edu

[†] Machine Learning Department
[‡] Department of Statistics and Data Science
Carnegie Mellon University
Pittsburgh, PA 15213

## Abstract

We study the problem of learning Ising models in a setting where some of the samples from the underlying distribution can be arbitrarily corrupted. In such a setup, we aim to design statistically optimal estimators in a high-dimensional scaling in which the number of nodes $p$, the number of edges $k$ and the maximal node degree $d$ are allowed to increase to infinity as a function of the sample size $n$. Our analysis is based on exploiting moments of the underlying distribution, coupled with novel reductions to univariate estimation. Our proposed estimators achieve an optimal dimension independent dependence on the fraction of corrupted data in the contaminated setting, while also simultaneously achieving high-probability error guarantees with optimal sample-complexity. We corroborate our theoretical results by simulations.

## 1 Introduction

Undirected graphical models (also known as Markov random fields (MRFs)) have gained significant attention as a tool for discovering and visualizing dependencies among covariates in multivariate data. Graphical models provide compact and structured representations of the joint distribution of multiple random variables using graphs that represent conditional independences between the individual random variables. They are used in domains as varied as natural language processing[38], image processing [9, 24, 27], spatial statistics [44] and computational biology [23], among others. Given samples drawn from the distribution, a key problem of interest is to recover the underlying dependencies represented by the graph. A slew of recent results [40, 43, 45] have shown that it is possible to learn such models even in domains and settings where the number of samples is potentially smaller than the number of variables. These results however make the common assumption that the sample data is clean, and have no corruptions. However, modern data sets that arise in various branches of science and engineering are no longer carefully curated. They are often collected in a decentralized and distributed fashion, and consequently are plagued with the complexities of outliers, and even adversarial manipulations.

Huber [28] proposed the $\epsilon$-contamination model as a framework to study such datasets with potentially arbitrary corruptions. In this setting, instead of observing samples directly from the true distribution $\mathbb{P}^\star$, we observe samples drawn from $\mathbb{P}_\epsilon$, which for an arbitrary distribution $Q$ is defined as a mixture

---

[*]Equal Contribution

model,

$$\mathbb{P}_\epsilon = (1 - \epsilon)\mathbb{P}^\star + \epsilon Q. \tag{1}$$

Then, given $n$ samples from $\mathbb{P}_\epsilon$, the goal is to recover functionals of $\mathbb{P}^\star$. There has been a lot of classical work on estimators for the $\epsilon$-contamination model setting that largely trade off computational versus statistical efficiency (see [29] and references therein). Moreover, there has been substantial progress [3, 7, 15, 16, 18, 32, 35, 42] on designing provably robust estimators which are computationally tractable while achieving near-optimal contamination dependence (*i.e.* dependence on the fraction of outliers $\epsilon$). However, to the best of our knowledge, there are no known results for learning general graphical models robustly.

## 1.1 Related Work

In this work, we focus on the specific undirected graphical model sub-class of Ising models [30]. There has been a lot of work for learning Ising models in the uncontaminated setting dating back to the classical work of Chow and Liu [8]. Csiszár and Talata [10] discuss pseudo likelihood based approaches for estimating the neighborhood at a given node in MRFs. Subsequently, a simple search based method is described in [6] with provable guarantees. Later, Ravikumar et al. [43] showed that under an incoherence assumption, node-wise (regularized) estimators provably recover the correct dependency graph with a small number of samples. Recently, there has been a flurry of work [5, 31, 37, 48, 50] to get computationally efficient estimators which recover the true graph structure without the incoherence assumption, including extensions to identity and independence testing [12]. However, all the aforementioned results are in the uncontaminated setting. Recently, Lindgren et al. [36] derived preliminary results for learning Ising models robustly. However, their upper and lower bounds do not match. Moreover, their analysis primarily focuses on the robustness of the Sparsitron algorithm in [31], and they do not explore the effect of the underlying graph and correlation structures comprehensively.

**Contributions.** In this work, we give the *first* statistically optimal estimator for learning Ising models under the $\epsilon$-contamination model. Our estimators achieve a dimension-independent asymptotic error as a function of the fraction of outliers $\epsilon$, while simultaneously achieving high probability deviation bounds. As an important special case of our results, we also close known sample complexity gaps in the uncontaminated setting for some classes of Ising models. We finally corroborate our theoretical findings with simulation studies.

## 1.2 Background and Problem Setup

We begin with some background on Ising models and then provide the precise formulation of the problem. We follow the notation of Santhanam and Wainwright [46] very closely.

Consider an undirected graph $G = (V, E)$ defined over a set of vertices $V = \{1, 2, \ldots, p\}$ with edges $E \subset \{(s, t) : s, t \in V, s \neq t\}$. The neighborhood of any node $s \in V$ is the subset $\mathcal{N}(s) \subset V$ given by $\mathcal{N}(s) \stackrel{\text{def}}{=} \{t | (s, t) \in E\}$, and the degree of any vertex $s$ is given by $d_s = |\mathcal{N}(s)|$. Then, the degree of a graph $d = \max_s d_s$ is the maximum vertex degree, and $k = |E|$ is the total number of edges. We obtain an MRF by associating a random variable $X_v$ at each vertex $v \in V$, and then considering a joint distribution $\mathbb{P}$ over the random vector $(X_1, \ldots, X_p)$. An Ising model is a special instantiation of an MRF where each random variable $X_s$ take values in $\{-1, +1\}$, and the joint probability mass function is given by:

$$\mathbb{P}_\theta(x_1, \ldots, x_p) \propto \exp\left(\sum_{1 \leq s < t \leq p} \theta_{st} x_s x_t\right), \tag{2}$$

where we view $\theta$ as the parameter vector of the distribution. Note that $\theta \in \mathbb{R}^{p \times p}$ is such that $\theta_{ij} = 0 \Leftrightarrow (i, j) \notin E$ and $\theta = \theta^T$.

**Graph Classes.** In this work, we consider two classes of Ising models (2) based on the conditions imposed on the edge set:

1. $\mathcal{G}_{p,d}$: the collection of graphs $G$ with $p$ vertices such that each vertex has at most $d$ neighbors for some $d \geq 1$, and

2. $\mathcal{G}_{p,k}$: the collection of graphs $G$ with $p$ vertices such that the total number of edges in the graph is at most $k$ for some $k \geq 1$.

In addition to these structural properties, we also consider some subclasses based on the parameters of the Ising model. We define the *model width* as:

$$\omega^*(\theta(G)) \overset{\text{def}}{=} \max_{u \in V} \sum_{v \in V} |\theta_{uv}|.$$

It is well-known (see for instance [46]) that estimation in Ising models becomes harder with increasing value of edge parameters, since, large values of edge parameters may hide the contributions of other edges. Similarly, we define the *minimum edge weight* as:

$$\lambda^*(\theta(G)) \overset{\text{def}}{=} \min_{(s,t) \in E} |\theta_{st}|.$$

With these structural and parameter properties in place, we define the classes of Ising models that we will be studying in the rest of the paper. Given a pair of positive numbers $(\lambda, \omega)$:

1. $\mathcal{G}_{p,d}(\lambda, \omega)$: the set of all Ising models defined over a graphs $G$ with $p$ vertices, with each vertex having degree at most $d$ and parameters satisfying

$$\lambda^*(\theta(G)) \geq \lambda \text{ and } \omega^*(\theta(G)) \leq \omega.$$

2. $\mathcal{G}_{p,k}(\lambda, \omega)$: the set of all Ising models defined over a graphs $G$ with $p$ vertices, with total number of edges at most $k$ and parameters satisfying

$$\lambda^*(\theta(G)) \geq \lambda \text{ and } \omega^*(\theta(G)) \leq \omega.$$

Furthermore, we work in the **high temperature regime** where we assume that the model width bound $\omega^*(\theta(G)) \leq 1 - \alpha$ for some $\alpha > 0$. Note that this assumption implies the Dobrushin condition [19], which in case of Ising models is given by

$$\max_{u \in V} \sum_{v \in V} \tanh(|\theta_{uv}|) \leq 1 - \alpha, \qquad \alpha \in (0, 1). \tag{3}$$

While this may seem restrictive, this assumption is widely popular for studying Ising models, for example, see related works in statistical physics [20, 47], mixing times of Glauber dynamics [13, 33], correlation decay [34] and more recently in estimation and testing problems [11, 12].

**Notation:** Given a matrix $M$ of dimensions $l \times m$, we will denote the $i^{th}$ row of matrix by $M_i$ and the $(i,j)^{th}$ element by $M_{ij}$ or $M(i,j)$. $M_{-i}$ denotes the sub-matrix formed by all rows except $i$, and analogously $M_{.,-j}$ denotes the sub-matrix formed by all columns except $j$. $M(i)$ denotes the vector $[M_i]_{-i}$ i.e., the $i^{th}$ row of $M$ excluding element $M_{ii}$. Given a vector $v$, $\|v\|_p = \sqrt[p]{\sum_i |v_i|^p}$ denotes its $\ell_p$-norm, and its $\ell_\infty$-norm is given by $\|v\|_{\max} = \max_i |v_i|$. For a matrix $M$, $\|M\|_{p,q}$ denotes the mixed $\ell_{p,q}$-norm, which is the $q$-norm of the collection of $p$-norms of the rows of $M$. We also use the shorthand $[d] = \{1, 2, \ldots, d\}$. We denote the total variation (TV) distance between two discrete distributions $p, q$ with support $\mathcal{X}$ by $d_{\text{TV}}(p, q) = \frac{1}{2} \sum_{x \in \mathcal{X}} |p(x) - q(x)|$.

## 2 Information-theoretic bounds for the $\epsilon$-contamination model

Recall that in the $\epsilon$-contamination model (1), we observe $n$ samples from $\mathbb{P}_\epsilon = (1 - \epsilon)\mathbb{P}^\star + \epsilon Q$. In this model, even in the asymptotic setting as $n \to \infty$, we cannot expect to recover the true parameters exactly. To see this, suppose that $\mathbb{P}_1^\star, \mathbb{P}_2^\star$ are such that there exist two distributions $Q_1$ and $Q_2$ such that

$$\mathbb{P}_\epsilon = (1 - \epsilon)\mathbb{P}_1^\star + \epsilon Q_1 = (1 - \epsilon)\mathbb{P}_2^\star + \epsilon Q_2,$$

then, we cannot hope to distinguish between the two distributions. It is easy to show (see [17]) that the above condition is equivalent to assuming that $d_{\text{TV}}(\mathbb{P}_1^\star, \mathbb{P}_2^\star) = \frac{\epsilon}{1-\epsilon}$. Thus, for any given contaminated distribution $\mathbb{P}_\epsilon$, there is a set of possible uncontaminated distributions (including the ground truth uncontaminated distribution among others) within a ball of some fixed radius with respect to the TV distance, any of which could give rise to the given contaminated distribution $\mathbb{P}_\epsilon$. Thus, when estimating the uncontaminated distribution with respect to some loss function, in the worst case we could incur loss corresponding to the farthest pair of distributions in the ball of some fixed radius with respect to TV distance. This is captured by the geometric notion of modulus of continuity [22], which can then be used to derive sharp bounds on estimation in such a setting:

**Definition 1** (TV modulus of continuity). *Given a loss function $L : \Theta \times \Theta \to \mathbb{R}^+$ defined over the parameter space $\Theta$, a class of distributions $\mathcal{D}$, a functional $f : \mathcal{D} \to \Theta$ and a proximity parameter $\epsilon$, the modulus of continuity $\omega(f, \mathcal{D}, L, \epsilon)$ is defined as*

$$\omega(f, \mathcal{D}, L, \epsilon) \stackrel{\text{def}}{=} \sup_{\substack{\mathbb{P}_1, \mathbb{P}_2 \in \mathcal{D} \\ d_{\text{TV}}(\mathbb{P}_1, \mathbb{P}_2) \leq \epsilon}} L(f(\mathbb{P}_1), f(\mathbb{P}_2)). \tag{4}$$

Intuitively, this quantity controls how far the functionals of two distributions can be, subject to the constraint that the TV distance between them is $\epsilon$. Note that for general Ising models, there do not exist *any* results that directly relate the total variation distance to the difference in parameters *i.e.* which study the TV modulus of continuity for the parameters of an Ising model.

A key contribution of our work is to establish sharp upper bounds on the TV modulus of continuity for parameter error in the high temperature regime. The loss function is considered to be the $\ell_{2,\infty}$ norm *i.e.* for matrices $x, y \in \mathbb{R}^{p \times p}$, $L(x, y) = \max_i \|x_i - y_i\|_2$.

**Theorem 1.** *Consider two Ising models defined over two graphs $G^{(1)}$ and $G^{(2)}$ with $p$ vertices with parameters $\theta^{(1)}$ and $\theta^{(2)}$ respectively, each of which satisfy the high temperature condition ([3]) with constant $\alpha$. If $d_{TV}\left(\mathbb{P}_{\theta^{(1)}}, \mathbb{P}_{\theta^{(2)}}\right) \leq \epsilon$, then we have that:*

$$\|\theta^{(1)}(i) - \theta^{(2)}(i)\|_2 \lesssim^1 \epsilon \sqrt{C_1(\alpha) \log\left(\frac{2}{\epsilon}\right)} \text{ for all } i \in [p],$$

*where $C_1(\alpha)$ is a constant depending on $\alpha$.*

Observe that Theorem [1] shows that the parameter error is *independent* of the dimension $p$, degree $d$ and the number of edges $k$. Furthermore, it is also independent of the minimum edge weight $\lambda$. As expected, when $\epsilon \to 0$, we see that the parameters are equal providing an alternate route to showing that the parameters of an Ising model are identifiable in the high temperature setting. We also establish that the dependence on $\epsilon$ is tight upto logarithmic factors by providing a complementary lower bound – proofs of which are made available in the appendix (Sections [C.1] and [C.2]).

**Lemma 1.** *There exists two Ising models satisfying the properties in Theorem [1] whose parameters $\theta^{(1)}$ and $\theta^{(2)}$ satisfy:*

$$\|\theta^{(1)}(i) - \theta^{(2)}(i)\|_2 \gtrsim \epsilon \text{ for all } i \in [p].$$

## 3 TV Projection Estimators

Recall the geometric picture of TV contamination discussed in the previous section: given the contaminated distribution, there is a set of possible uncontaminated distributions within a ball of some fixed radius with respect to TV. It is thus natural to consider the TV projection of the contaminated distribution onto the set of all possible uncontaminated distributions. These are also called *minimum distance estimators* and were proposed by Donoho and Liu [21], which we consider for our setting to learn Ising models robustly, leveraging our Theorem [1].

### 3.1 Population Robust Estimators for $\mathcal{G}_p$

Let us first consider the population setting *i.e.*, in which we have distribution access to the contaminated distribution $\mathbb{P}_\epsilon = (1 - \epsilon)\mathbb{P}_{\theta^\star} + \epsilon Q$, where $\mathbb{P}_{\theta^\star} \in \mathcal{G}_p(\lambda, \omega)$ [2]. In this setting, we use the minimum distance estimator [21] to construct robust estimators. In particular, let $\mathbb{P}_{\widehat{\theta}_{\text{MDE}}}$ be the minimum distance estimate defined as

$$\mathbb{P}_{\widehat{\theta}_{\text{MDE}}} = \operatorname*{argmin}_{\mathbb{P}_\theta \in \mathcal{G}_p} d_{\text{TV}}(\mathbb{P}_\theta, \mathbb{P}_\epsilon). \tag{5}$$

This estimator is effectively the TV projection of the contaminated distribution onto the set of all Ising model distributions whose underlying graph lies in $\mathcal{G}_p$.

Noting that $d_{\mathrm{TV}}(\mathbb{P}_{\theta^\star}, \mathbb{P}_{\widehat{\theta}_{\mathrm{MDE}}}) \leq \epsilon$, by an application of the triangle inequality we have that $d_{\mathrm{TV}}(\mathbb{P}_{\theta^\star}, \mathbb{P}_{\widehat{\theta}_{\mathrm{MDE}}}) \leq 2\epsilon$. Combining this with Theorem 1, we get that,

$$\|\widehat{\theta}_{\mathrm{MDE}}(i) - \theta^\star(i)\|_2 \lesssim \epsilon\sqrt{C(\alpha)\log\left(\frac{2}{\epsilon}\right)} \quad \text{for all } i \in [p].$$

**Corollary 1.** *Let $\mathbb{P}_{\widehat{\theta}_{\mathrm{MDE},\lambda}}$ be the TV projection of the contaminated distribution $\mathbb{P}_\epsilon$ onto the class of Ising models $\mathcal{G}_{p,d}$ with minimum edge weight at least $\lambda$. Define the edge set of $\mathbb{P}_{\widehat{\theta}_{\mathrm{MDE}}}$ as $E(\widehat{\theta}_{\mathrm{MDE},\lambda}) = \{(i,j) : |\widehat{\theta}_{\mathrm{MDE},\lambda}(i,j)| > \frac{\lambda}{2}\}$. When $\epsilon\sqrt{C(\alpha)\log\left(\frac{2}{\epsilon}\right)} \leq \frac{\lambda}{2C_1}$, where $C_1$ is a universal constant, the edge sets of $\mathbb{P}_{\widehat{\theta}_{\mathrm{MDE},\lambda}}$ and $\mathbb{P}_{\theta^\star}$ coincide i.e.,*

$$E(\widehat{\theta}_{\mathrm{MDE},\lambda}) = E(\theta^\star).$$

Observe that this result is interesting and surprising, because one would generally not expect to be able to recover the true edge $E(\theta^\star)$ under contamination. Additionally, as mentioned earlier, there is no dependence on $p$, $d$ or $k$, which means that the irrespective of the size of graph, if the minimum edge weight is sufficiently large or the level of contamination is sufficiently small, we would be able to recover the true edge set in the infinite sample limit.

## 3.2 Empirical Robust Estimators for $\mathcal{G}_{p,k}$

The minimum distance estimator is not suitable for non-asymptotic settings since we do not have access to the population contaminated distribution, but only to its discrete empirical counterpart, obtained via samples from the contaminated distribution. It would thus be ideal if there were an approximation to the TV distance that is amenable to projections of discrete distributions, and that preserves the optimality properties of the full TV projections.

Remarkably, Yatracos [51] proposed just such an approximation to TV projections. Consider a class of distributions $\mathcal{P}$. It is known that $d_{\mathrm{TV}}(P,Q) = \sup_A |P(A) - Q(A)|$, where the supremum is over all possible measurable sets $A \subseteq \mathrm{supp}(P)$. While uniform convergence fails over all sets, Yatracos [51] showed that we can consider a much smaller collection of clevely chosen sets. In particular, Yatracos [51] suggested approximating the TV distance between distribution $P, Q \in \mathcal{P}$ as

$$d_{\mathrm{TV}}(P,Q) \approx \sup_{A \in \mathcal{A}} |P(A) - Q(A)|,$$

where $\mathcal{A}$ are sets of the form

$$\mathcal{A} = \{A(\mathbb{P}_1, \mathbb{P}_2) : \mathbb{P}_1, \mathbb{P}_2 \in \mathcal{P}\}, \tag{6}$$

and $A(\mathbb{P}_1, \mathbb{P}_2) = \{x : \mathbb{P}_1(x) > \mathbb{P}_2(x)\}$. This approximation allows us to construct statistically optimal estimators for $\mathcal{G}_{p,k}$.

### 3.2.1 Non-Asymptotic Robust Estimators for $\mathcal{G}_{p,k}$

Given samples $\{x^{(i)}\}_{i=1}^n$ from the mixture model $\mathbb{P}_\epsilon$ defined in (1), define $\widehat{\mathbb{P}}_{n,\epsilon}(A) = \frac{1}{n}\sum_{i=1}^n \mathbb{I}\{x^{(i)} \in A\}$ for all $A \in \mathcal{A}$, where $\mathcal{A}$ is the same as defined in (6) with the class of distributions $\mathcal{G}_{p,k}$. Our estimator is defined as

$$\mathbb{P}_{\widehat{\theta}} = \operatorname*{argmin}_{\mathbb{P}_\theta \in \mathcal{G}_{p,k}} \sup_{A \in \mathcal{A}} \left|\mathbb{P}_\theta(A) - \widehat{\mathbb{P}}_{n,\epsilon}(A)\right|. \tag{7}$$

The following lemma characterizes the performance of our estimator.

**Lemma 2.** *Given $n$ samples from a contaminated distribution $P_\epsilon$, the Yatracos estimate (7) satisfies with probability least $1 - \delta$:*

$$d_{\mathrm{TV}}(\mathbb{P}_{\widehat{\theta}}, \mathbb{P}_{\theta^\star}) \leq 2\epsilon + \mathcal{O}\left(\sqrt{\frac{k\log(p^2 e/k)}{n}} + \sqrt{\frac{\log(1/\delta)}{n}}\right).$$

The lemma above shows that the Yatracos estimate is close to the true Ising model in TV distance with high-probability. Combining Lemma 2 and Theorem 1, we get parameter error guarantees for the Yatracos estimate.

**Corollary 2.** *Given $n$ samples from $\mathbb{P}_\epsilon$, the Yatracos' estimator returns a $\widehat{\theta}$ such that with probability at least $1 - \delta$,*

$$\|\widehat{\theta}(i) - \theta^\star(i)\|_2 \lesssim 2\epsilon\sqrt{\log(1/\epsilon)} + \widetilde{\mathcal{O}}\left(\sqrt{\frac{k\log(p^2 e/k)}{n}} + \sqrt{\frac{\log(1/\delta)}{n}}\right) \quad \text{for all } i \in [p], \quad (8)$$

*where $\tilde{\mathcal{O}}(.)$ hides logarithmic factors involving its argument.*

**Remarks.** Note that the proposed estimator achieves the same (asymptotic) dimension-independent error as the Minimum Distance Estimate discussed in Section 3.1, while simultaneously achieving an $\widetilde{\mathcal{O}}\left(\sqrt{\frac{k\log p}{n}}\right)$ error rate. Moreover, observe that in the uncontaminated setting, i.e., when $\epsilon = 0$, this is the *first* estimator to get an $\widetilde{O}\left(\sqrt{\frac{k\log p}{n}}\right)$ error rate. As a consequence, Yatracos' estimator followed by an additional thresholding step gives the first estimator to recover the true edge set $E(\theta^\star)$ with only $\widetilde{\mathcal{O}}\left(\frac{k\log(p)}{\lambda^2}\right)$ samples. In contrast, the estimator proposed by [46] posit that the sample size should satisfy $\mathcal{O}(1/\lambda^4)$ when the parameters are unknown. In the contaminated case, note that we show a better dependence on $\epsilon - \mathcal{O}(\epsilon\sqrt{\log(1/\epsilon)})$ vs. $\sqrt{\epsilon}$ in [36]. The proof for Lemma 2 is presented in Section D.2 of the appendix. A similar analysis was conducted in [14], however [14] study density estimation, and not parameter estimation. The bound on the modulus of continuity obtained in Theorem 1 allows us to relate the TV distance between the estimated distribution and the true distribution to the parameter error, thus giving us bounds for parameter estimation.

### 3.2.2 Non-Asymptotic Robust Estimators for $\mathcal{G}_{p,d}$

Under the same setting as considered for $\mathcal{G}_{p,k}$, we see that directly employing the estimator (7) would lead to a sub-optimal rate. Our guarantee for (7) for $\mathcal{G}_{p,k}$ relies on the fact that parameters for Ising models in $\mathcal{G}_{p,k}$ contain at most $k$ non-zero elements, hence the subset $A(\theta^{(1)}, \theta^{(2)}) = \{x : \mathbb{P}_{\theta^{(1)}}(x) > \mathbb{P}_{\theta^{(2)}}(x)\}$ is a half-space defined by a vector with at most $2k + 1$ non-zero elements. However, these subsets defined with parameters $\theta^{(1)}, \theta^{(2)}$ of two Ising models in $\mathcal{G}_{p,d}$ is a half-space defined by a vector that have at most $pd + 1$ non-zero elements. This leads to a rate term that is proportional to $\sqrt{pd\log(p)/n}$, which does not scale well in high-dimensional settings.

## 4  Robust Conditional Likelihood Estimators

In the previous section, we have seen that the estimator based on Yatracos classes [51] provides an approximate TV projection for $\mathcal{G}_{p,k}$ but not for $\mathcal{G}_{p,d}$. The main caveat with this estimator is that it is not tractable and takes infinite time. To circumvent this issue, we consider a more direct approach to robust estimation: we "robustify" the gradient samples obtained from samples $\{x^{(i)}\}_{i=1}^n$ of the contaminated distribution $\mathbb{P}_\epsilon = (1 - \epsilon)\mathbb{P}_{\theta^\star} + \epsilon Q$.

**Neighborhood-based logistic regression.** In a classical paper, Besag [4] made the key structural observation that under model (2), the conditional distribution of node $X_i$ given the other variables $X_{-i} = x_{-i}$ is given by

$$\mathbb{P}_{\theta^\star}(X_i = x_i | X_{-i} = x_{-i}) = \frac{\exp(2x_i \sum_{t \in \mathcal{N}(i)} \theta_{it}^\star x_t)}{\exp(2x_i \sum_{t \in \mathcal{N}(i)} \theta_{it}^\star x_t) + 1} = \sigma(x_i \langle 2\theta^\star(i), x_{-i}\rangle).$$

Thus the variable $X_i$ can be viewed as the response variable in a logistic regression model with $X_{-i}$ as the covariates and $2\theta^\star(i)$ as the regression vector. In particular, this implies that $\mathbb{E}_{x \sim \mathbb{P}_{\theta^\star}}[\nabla l_i(2\theta^\star(i); x)] = \mathbf{0}$ where $\ell_i(\theta(i); x) = \log\sigma(x_i \langle \theta(i), x_{-i}\rangle)$ is the conditional log-likelihood of $x$ under $\mathbb{P}_\theta$. Note that for graphs with maximum degree at most $d$, the parameter vector $\theta^\star(i)$ has at most $d$ non-zero entries, and for graphs with at most $k$ edges, the parameter vector $\theta^\star(i)$ has at most $k$ non-zero entries. Ravikumar et al. [43] solved an $\ell_1$-regularized logistic regression to recover the node parameters for graphs with bounded maximum degree. However, in our setting, the data is contaminated with outliers, and hence the minimizer of the likelihood can be arbitrarily bad. While there has been recent work giving provably optimal algorithms for robust logistic regression [42], all of these results are in the low-dimensional setting. We propose the *first* statistically optimal estimator for sparse logistic regression, and use that to provide estimators for learning Ising models.

---

**Algorithm 1** Robust1DMean - Robust univariate mean estimator

---

**Require:** Samples $\{z^{(i)}\}_{i=1}^{2n}$, corruption level $\epsilon$, confidence level $\delta$

1: Split $\{z^{(i)}\}_{i=1}^{2n}$ into two subsets $\mathcal{Z}_1 = \{z^{(i)}\}_{i=1}^{n}$ and $\mathcal{Z}_2 = \{z^{(i)}\}_{i=n+1}^{2n}$

2: Set $\beta = \max\left(\epsilon, \frac{\log(1/\delta)}{n}\right)$

3: $n_1 = n\left(1 - 2\beta - \sqrt{2\beta \log(4/\delta)/n} - \log(4/\delta)/n\right)$

4: Using $\mathcal{Z}_1$, identify $\widehat{I} = [a, b]$ which is the shortest interval containing $n_1$ points

5: **return** $\frac{1}{n_2}\sum\limits_{i=n+1}^{2n} z^{(i)}\mathbb{I}\left\{z^{(i)} \in \widehat{I}\right\}$ where $n_2 = \sum\limits_{i=n+1}^{2n} \mathbb{I}\left\{z^{(i)} \in \widehat{I}\right\}$

---

**Robust Sparse Logistic Regression.** Our approach is based on a reduction to robust univariate estimation initially proposed by [41]. In particular, note that when we have clean data, then, in the population setting, $\theta^\star(i)$ is the unique solution to the equation $\|\mathbb{E}_{x\sim\mathbb{P}_{\theta^\star}}\left[\nabla\ell_i(\theta(i); x)\right]\|_2 = \mathbf{0}$ or equivalently, it is the unique minimizer for the following optimization problem:

$$\theta^\star(i) = \operatorname*{argmin}_{w:\|w\|_0 \leq s} \sup_{u \in \mathcal{S}^{p-2}} \left|\mathbb{E}_{x\sim\mathbb{P}_{\theta^\star}}[u^T\nabla\ell_i(w; x)]\right|,$$

where we have simply used the variational form of the norm of a vector. Observe that $\mathbb{E}_{x\sim\mathbb{P}_{\theta^\star}}[u^T\nabla\ell_i(w; x)]$ is simply the population (uncontaminated) mean of the gradients, when projected along the direction $u$. Unfortunately, we only have finite samples which are moreover contaminated. We can pass these univariate projections of the gradient through a *robust* univariate mean estimator, and return a point which has the *smallest* (robust) mean along any direction. This leads to the following program,

$$\widehat{\theta}(i) = \operatorname*{argmin}_{w \in \mathcal{N}_s^\gamma(\mathcal{S}^{p-2})} \sup_{u \in \mathcal{N}_{2s}^{1/2}(\mathcal{S}^{p-2})} \left|\mathsf{Robust1DMean}(\{u^T\nabla\ell_i(w; x^{(j)})\}_{j=1}^n)\right|, \tag{9}$$

where $\mathcal{N}_s^\gamma(\mathcal{S}^{p-2})$ is a $\gamma$-cover of the unit sphere over $p-1$ dimensions with $s$ non-zero entries i.e., for every $x \in \mathcal{S}^{p-2}$ that has $s$ non-zero entries, there exists $y \in \mathcal{N}_s^\gamma(\mathcal{S}^{p-2})$ such that $\|x - y\|_2 \leq \gamma$. Our robust univariate mean estimator is based on the shortest interval estimator (Shorth) studied in [2, 35, 41]. The estimator, presented in Algorithm 1, proceeds by using half of the samples to identify the shortest interval containing roughly $(1-\epsilon)n$ fraction of the points, and then the remaining half of the points is used to return an estimate of the mean. Intuitively, this estimator effectively trims distant outliers, thereby limiting their influence on the estimate.

We assume that the contamination level $\epsilon$, confidence parameter $\delta$, and sparsity $s$ are such that,

$$2\epsilon + \sqrt{\epsilon\left(\frac{s\log(p)}{n} + \frac{\log(p/\delta)}{n}\right)} + \frac{s\log(p)}{n} + \frac{\log(4p/\delta)}{n} < c, \tag{10}$$

for some small constant $c > 0$. As noted earlier, the sparsity parameter $s$ is the maximum degree $d$ for $\mathcal{G}_{p,d}$ and the maximum number of edges $k$ for $\mathcal{G}_{p,k}$.

**Theorem 2** (Guarantees for $\mathcal{G}_{p,d}$)**.** *Under the setting considered in 4 along with Assumption (3), the estimator in (9) returns estimates $\{\widehat{\theta}(i)\}_{i=1}^p$ with $\gamma = \max\left\{\frac{\epsilon}{p}, \frac{\log(1/\delta)}{np}\right\}$ returns with probability at least $1 - \delta$*

$$\|\widehat{\theta}(i) - \theta^\star(i)\|_2 \lesssim \epsilon\sqrt{C(\alpha)\log\left(\frac{1}{\epsilon}\right)} + \sqrt{C(\alpha)\frac{d}{n}\log\left(\frac{3ep^2}{d\gamma}\right)} + \max\left(\epsilon, \frac{\log(1/\delta)}{n}\right) \text{ for all } i \in [p].$$

**Corollary 3** (Guarantees for $\mathcal{G}_{p,k}$)**.** *Under the setup considered in Theorem 2, the estimator in (9) returns estimates $\{\widehat{\theta}(i)\}_{i=1}^p$ with $\gamma = \max\left\{\frac{\epsilon}{p}, \frac{\log(1/\delta)}{np}\right\}$ returns with probability at least $1 - \delta$*

$$\|\widehat{\theta}(i) - \theta^\star(i)\|_2 \lesssim \epsilon\sqrt{C(\alpha)\log\left(\frac{1}{\epsilon}\right)} + \sqrt{C(\alpha)\frac{k}{n}\log\left(\frac{3ep^2}{k\gamma}\right)} + \max\left(\epsilon, \frac{\log(1/\delta)}{n}\right) \text{ for all } i \in [p].$$

**Remarks.** Observe that our estimator achieves the same (asymptotic) bias as the Minimum Distance Estimator, previously discussed in Section 3.1. Define the recovered edge set as those edges $(i, j)$

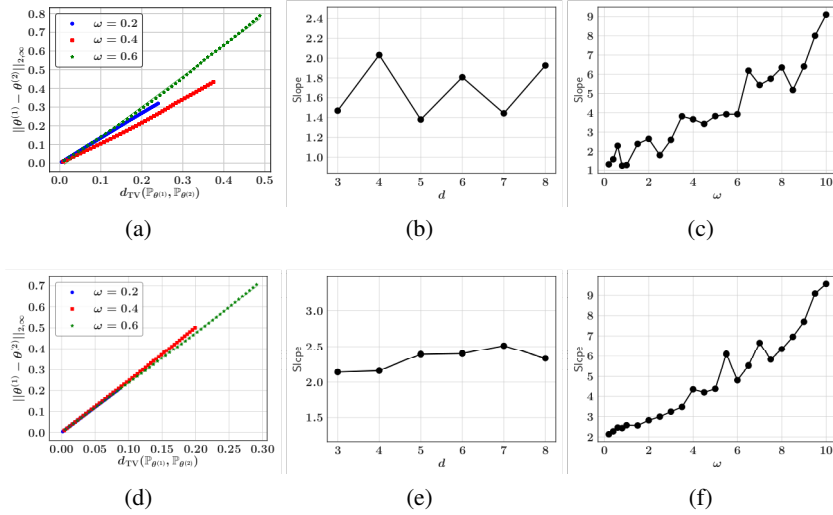

Figure 1: Left: Variation of $\|\theta^{(1)} - \theta^{(2)}\|_{2,\infty}$ with $d_{\text{TV}}(\mathbb{P}_{\theta^{(1)}}, \mathbb{P}_{\theta^{(2)}})$ for $G^{(1)}, G^{(2)} \in \mathcal{G}_{15,4}^{\text{clique}}$ (top) and $G^{(1)}, G^{(2)} \in \mathcal{G}_{15,4}^{\text{star}}$ (bottom) graphs with varying $\omega$. Middle: Variation of slope with $d$ for cliques (top) and star (bottom) with $p = 12$ and $\omega = 0.4$. Right: Variation of slope with $\omega$ for cliques (top) and star (bottom) with $p = 15$ and $d = 5$. The slope is defined as $\frac{\|\theta^{(1)} - \theta^{(2)}\|_{2,\infty}}{d_{\text{TV}}(\mathbb{P}_{\theta^{(1)}}, \mathbb{P}_{\theta^{(2)}})}$.

satisfying $|\widehat{\theta}_{ij}| \geq \lambda/2$. When $\epsilon = 0$, i.e., no contamination, for $\mathcal{G}_{p,d}$, we require the number of samples $n \geq \mathcal{O}\left(\frac{d\log(p)}{\lambda^2}\right)$ to recover the true edge set $E(\theta^\star)$. Even in the uncontaminated setting, there is *no* known estimator which achieves the same optimal sample complexity as ours. In particular, Santhanam and Wainwright [46] achieve similar rates when they assume that the structure is known, while other approaches of [37, 43] have worse dependence on the degree $d$. Hence, our proposed estimator has an optimal (asymptotic) bias and optimal high probability bounds. For $\mathcal{G}_{p,k}$, we obtain the same rate and sample complexity as Yatracos' estimator (7), which we remarked is optimal. The proof of Theorem 2 is presented in Section E.1 of the appendix.

## 5 Synthetic Experiments

Our theoretical results crucially hinge on bounds on the TV modulus of continuity derived in Theorem 1, and we devote this section to corroborating these bounds.

**Setup.** We consider two different ensembles. A graph $G \in \mathcal{G}_{p,d}^{\text{star}}$ when one of the $p$ nodes is connected $d$ other vertices, and no other edges are present in the graph, resembling a star. A graph $G \in \mathcal{G}_{p,d}^{\text{clique}}$ contains $\lfloor\frac{p}{d+1}\rfloor$ cliques of size $d+1$, and the remainder of the nodes $p \mod (d+1)$ fully connected amongst themselves. We generate our plots in the following manner: first we construct two graphs with the same structure - either from $\mathcal{G}_{p,d}^{\text{clique}}$ of $\mathcal{G}_{p,d}^{\text{star}}$. We instantiate parameters for the first graph with $\theta^{(1)}$ with model width $\omega$ and then vary the parameters for the second graph as $\theta^{(2)} = \theta^{(1)} \cdot \frac{i}{25}$ for $i$ ranging from 1 to 50. We vary $p \in \{12, 15\}$, $d \in \{3 : 8 : 1\}$ and $\omega \in \{0.2 : 1.0 : 0.2\} \cup \{1.5 : 10 : 0.5\}$ where $\{a : b : c\}$ denotes values between $a$ and $b$ (both inclusive) with consecutive values differing by $c$.

**Results.** Figures 1(a) and 1(d) exhibits a linear relationship between $d_{\text{TV}}(\mathbb{P}_{\theta^{(1)}}, \mathbb{P}_{\theta^{(2)}})$ and $\|\theta^{(1)} - \theta^{(2)}\|_{2,\infty}$, as suggested by our theoretical results from previous sections. Furthermore, we notice that the slope is not drastically affected by $\omega$, which also suggests that the constant $C(\alpha)$ appearing in our results is $O(1)$. We also note from Figures 1(b) and 1(e), that the slope is unaffected by a change in degree. Finally, in Figures 1(c) and 1(f), we notice the variation in the slope with increasing model width $\omega$. While our current result study the case when $\omega < 1$, it is also interesting to note an increasing trend when $\omega \geq 1$ suggesting an explicit dependence on $\omega$ in the low-temperature regime.

# 6 Discussion and Future Work

In this work we provided the first statistically optimal robust estimators for learning Ising models in the high temperature regime. Our estimators achieved optimal asymptotic error in the $\epsilon$-contamination model, and also high-probability deviation bounds in the uncontaminated setting. There are several avenues for future work, some of which we discuss below.

**Beyond Dobrushin's conditions.** In the low-temperature setting, Lindgren et al. [36] showed the existence of an estimator which gets an $O(\sqrt{\epsilon})$ error. In Appendix A, we tighten this for edge-bounded graphs by providing estimators which achieve $O(\min(\sqrt{\epsilon}, \epsilon\sqrt{k}))$ error, where $k$ is the maximum number of edges in the graph. However, giving matching lower bounds in this setting is an open problem. Our synthetic experiments surprisingly show that one may expect similar rates in the two temperature regimes.

**Computationally Efficient Estimators.** While in this work, we designed statistically optimal estimators that achieve an $O(\epsilon\sqrt{\log(1/\epsilon)})$ parameter error, whereas, existing computationally efficient approaches [31, 36] achieve a sub-optimal error of $O(\sqrt{\epsilon})$. Developing computationally efficient algorithms which close this gap is an interesting open problem.

**Other Contamination Models.** In this work, our focus was on designing estimators for the $\epsilon$-contaminated model, i.e., where a fraction of the data is arbitrarily corrupted. Another model of corruption - motivated by sensor networks and distributed computation where node failures are common - is when only a few features(nodes) get corrupted, and we still want to learn the appropriate graph structure for the uncontaminated nodes. Recent work by Goel et al. [25] discusses results for this model of contamination.

## Broader Impact

In this work, we provide statistically optimal estimators for learning Ising models under contamination. Ising models are themselves used in a variety of domains to learning relationship between pairs of binary random variables. One extremely interesting application is in the field of opinion analysis and voting network analysis. For instance, the nodes of the graph represent the voting base and the samples given to us are votes made of a series of topics as obtained via polls. Such estimators will help capture associations between voters. However, in a day and age where voting patterns are susceptible to adversarial corruptions, it is safe to assume that the vote samples are corrupted too. Using standard methods such as $\ell_1$-regularized logistic regression could have the unintended consequence of amplifying the biases from corrupted data, leading to poor judgements, whereas our methods are optimal resilient to such corruptions. However, if used without prior analysis of the data presented, this could potentially reduce the effect of outlier samples, which in the case of voting patterns, are representative of a minority groups.

## Acknowledgements

AP, VS and PR acknowledge the support of NSF via IIS-1955532, OAC-1934584, DARPA via HR00112020006, and ONR via N000141812861. SB and AP acknowledge the support of NSF via DMS-17130003 and CCF-1763734. We would also like to thank an anonymous reviewer for pointing out related work by Devroye et al. [14].

## Footnotes

[1]Here and throughout our paper we use the notation $\lesssim$ to denote an inequality with universal constants dropped for conciseness.

[2]We define the class $\mathcal{G}_p(\lambda, \omega)$ as the set of Ising models defined over $p$ vertices with minimum edge weight $\lambda$ and model width $\omega$

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
