[Supplementary Material]

# A  Beyond Dobrushin's Conditions.

All of our previous results are under the high temperature condition (3), where we rely of special properties of Ising models namely sub-Gaussianity of Ising models random variables. Following this effort, we attempt to analyze classes of Ising models where this condition doesn't hold to present an even more general analysis. Towards this end, we present moduli of continuity bounds as presented in Theorem 1. Here, we look out for dependence in the model width parameter in addition to the effective dimensionality of the problem ($d$ in the case of $\mathcal{G}_{p,d}$ and $k$ in the case of $\mathcal{G}_{p,k}$, and the tolerance parameter $\epsilon$.

**Theorem 3.** *Consider two Ising models defined over two graphs $G^{(1)}$ and $G^{(2)}$ over $p$ vertices with parameters $\theta^{(1)}$ and $\theta^{(2)}$ respectively, satisfying $\omega(\theta^{(1)}), \omega(\theta^{(2)}) \leq \omega$. If $d_{\mathrm{TV}}(\mathbb{P}_{\theta^{(1)}}, \mathbb{P}_{\theta^{(2)}}) \leq \epsilon$, then we have the following results for all $i \in [p]$:*

*(a) If $G^{(1)}, G^{(2)} \in \mathcal{G}_{p,d}$, then*

$$\|\theta^{(2)}(i) - \theta^{(1)}(i)\|_2 \lesssim \min\{\sqrt{\epsilon}, \epsilon\sqrt{d}\}\, \omega \exp(O(\omega)). \tag{11a}$$

*(b) If $G_1, G_2 \in \mathcal{G}_{p,k}$, then*

$$\|\theta^{(2)}(i) - \theta^{(1)}(i)\|_2 \lesssim \min\{\sqrt{\epsilon}, \epsilon\sqrt{k}\}\, \omega \exp(O(\omega)). \tag{11b}$$

Similar to Theorem 3, we get a modulus of continuity bound for the loss function defined by the $(2, \infty)$-norm. Note that as $\epsilon$ tends to 0, the bounds also tend to 0. However, it is worth noting that our primitive analysis contains an additional factor in $d/k$ based on the graph class considered. The sub-optimality is clear when we set $\omega = O(1)$, and the bounds while retaining a optimal dependence on $\epsilon$ have an additional dependence with $d/k$ when compared to the result in Theorem 1. Our analysis of the Yatracos estimator (7) does not depend of any specific bounds on the model width, and hence with the derived modulus of continuity bound, we arrive at the following corollary for the estimation error of the Yatracos estimate:

**Corollary 4.** *Given $n$ samples from the distribution $\mathbb{P}_\epsilon = (1 - \epsilon)\mathbb{P}_{\theta^\star} + \epsilon Q$, where $\mathbb{P}_{\theta^\star} \in \mathcal{G}_{p,k}(\lambda, \omega)$ and $Q$ is an arbitrary distribution supported over $\{-1, +1\}^p$, the parameter of Yatracos estimate (7) satisfies:*

$$\|\widehat{\theta}(i) - \theta^\star(i)\|_2 \lesssim \sqrt{k}\omega e^{\mathcal{O}(\omega)}\epsilon + \mathcal{O}\left(k\omega e^{\mathcal{O}(\omega)}\sqrt{\frac{\log(p^2 e/k)}{n}} + \sqrt{\frac{\log(1/\delta)}{n}}\right) \quad \text{for all } i \in [p].$$

Note that as $n \to \infty$, the bias of the estimator has optimal dependence on $\epsilon$, but incurs an additional dependence of $\sqrt{k}$. For $\epsilon = 0$ *i.e.* no contamination, the rate we achieve is approximately $\omega e^{\mathcal{O}(\omega)}k\sqrt{\frac{\log(p)}{n}}$, which leads to the number of samples $n \geq \mathcal{O}\left(\frac{k^2\omega^2 e^{\mathcal{O}(\omega)}\log(p)}{\lambda^2}\right)$ required to recover the true edge set $E(\theta^\star)$, and this is comparable to existing sample complexity results for learning Ising models belonging to $\mathcal{G}_{p,k}(\lambda, \omega)$ [46]. We present the proof of Theorem 3 in Section F.

# B Useful Properties of Ising models

In this section, we summarize some useful properties of Ising models which we use judiciously in our proofs. These results have appeared in previous work, but we state them for the sake of completeness.

## B.1 Sub-Gaussianity of Ising model distributions in the high temperature regime

First, we present a result from [26], which states that a random variable distributed according to an Ising model in the high temperature regime is sub-Gaussian.

**Proposition 3** ([26, Theorem 1.4]). *Let $z \sim \mathbb{P}$ be a random variable whose distribution $\mathbb{P}$ is an Ising model over $p$ nodes in the high temperature regime ([3](3)) with constant $\alpha$. Then for $v \sim \mathbb{R}^p$:*

$$\Pr_{z \sim \mathbb{P}} \left( |\langle v, z \rangle| > t \right) \leq 2 \exp \left( -\frac{t^2}{C(\alpha)||v||_2^2} \right), \tag{12}$$

*where $C(\alpha)$ is a constant depending on $\alpha$.*

## B.2 Strong convexity of the negative conditional log-likelihood

Here we present a proposition that states that the population negative conditional log-likelihood is strongly convex. This proposition is obtained using a result by Dagan et al. [11]. We first state the result by Dagan et al. [11] below, and then use it to show that the population negative condition log-likelihood is strongly convex.

**Proposition 4** ([11, Lemma 10]). *Let $z$ be a random variable distributed w.r.t. an Ising model over $p$ nodes whose parameter $\theta$ satisfies $\max_{i \in [p]} \|\theta(i)\|_\infty \leq \omega$ and $\min_{i \in [p]} \mathbb{P}_\theta(X_i = 1 | X_{-i} = x_{-i})(1 - \mathbb{P}_\theta(X_i = 1 | X_{-i} = x_{-i})) \geq \gamma$. Then for any $v \in \mathbb{R}^p$, we have that:*

$$\mathrm{Var}[\langle v, z \rangle] \geq \frac{C_1 \gamma^2 ||v||_2^2}{\omega},$$

*where $C_1$ is a universal constant.*

Now, let $\mathcal{L}_{\theta,i}(w)$ be the population negative conditional log-likelihood for node $X_i$, where $X$ is sampled from the Ising model distribution $\mathbb{P}_\theta$. Formally, $\mathcal{L}_{\theta,i}(w) = -\mathbb{E}_{z \sim \mathbb{P}_\theta}[\ell_i(w; z)]$, where $\ell_i(w; z)$ is the conditional log-likelihood of $z$ under $\mathbb{P}_\theta$ with respect to the $i^{th}$ node. As stated earlier, by the maximum likelihood principle, $\nabla \mathcal{L}_{\theta,i}(2\theta(i)) = \mathbf{0}$. With this definition, we have the Hessian of the population negative conditional log-likelihood as $\nabla^2 \mathcal{L}_{\theta,i}(w) = \mathbb{E}_{z \sim \mathbb{P}_\theta}[\nabla^2 \ell_i(w; z)]$. Then, we have the following result.

**Proposition 5.** *Let $\mathbb{P}_\theta$ be an Ising model over $p$ nodes whose parameter satisfies $\max_{i \in [p]} \|\theta(i)\|_\infty \leq \omega$, and let $w \in \mathbb{R}^{p-1}$ be such that $\|w\|_1 \leq 2\omega$. Then, for any vector $v \in \mathbb{R}^{p-1}$, there exists a universal constant $C > 0$ such that:*

$$v^T \nabla^2 \mathcal{L}_{\theta,i}(w) v \geq C \frac{\exp(-O(\omega))}{\omega} \|v\|_2^2.$$

*Proof.* First, observe that

$$\nabla^2 \mathcal{L}_{\theta,i}(w) = \mathbb{E}_{z \sim \mathbb{P}_\theta} \left[ \sigma(z_i \langle w, z_{-i} \rangle)(1 - \sigma(z_i \langle w, z_{-i} \rangle)) z_{-i} z_{-i}^T \right]$$
$$\Rightarrow v^T \nabla^2 \mathcal{L}_{\theta,i}(w) v = \mathbb{E}_{z \sim \mathbb{P}_\theta} \left[ \sigma(z_i \langle w, z_{-i} \rangle)(1 - \sigma(z_i \langle w, z_{-i} \rangle)) \langle z_{-i}, v \rangle^2 \right].$$

In Lemma [6](6), we show that for any $\|w\|_1 \leq 2\omega$, we have that

$$\sigma(z_i \langle w, z_{-i} \rangle)(1 - \sigma(z_i \langle w, z_{-i} \rangle)) \geq \frac{\exp(-2\omega)}{4}. \tag{13}$$

We now lower bound $\mathbb{E}[\langle z_{-i}, v \rangle^2]$. Since Ising model has zero mean field, we have that $\mathbb{E}[\langle z_{-i}, v \rangle^2] = \mathrm{Var}[\langle z_{-i}, v \rangle]$. Furthermore, due the assumptions placed on the parameter of the Ising model, we

obtain that for any $x \in \{-1, +1\}^{p-1}$, $\mathbb{P}_\theta(X_i = 1|X_{-i} = x)(1 - \mathbb{P}_\theta(X_i = 1|X_{-i} = x)) \geq \frac{1}{4}\exp(-4\omega)$. This can be shown as follows. For any $z \in \{-1, +1\}$ and $x \in \{-1, +1\}^{p-1}$, we have that:

$$\mathbb{P}_\theta(X_i = z|X_{-i} = x) = \frac{1}{1 + \exp(-z \langle 2\theta(i, -i), x \rangle)}$$

$$\overset{(i)}{\geq} \frac{1}{1 + \exp(2\omega)}$$

$$\geq \frac{1}{2\exp(2\omega)} = \frac{\exp(-2\omega)}{2}$$

$$\Rightarrow \mathbb{P}_\theta(X_i = 1|X_{-i} = x)\mathbb{P}_\theta(X_i = 0|X_{-i} = x) \geq \frac{\exp(-2\omega)}{2}\frac{\exp(-2\omega)}{2}$$

$$= \frac{\exp(-4\omega)}{4}$$

where Step $(i)$ uses Hölder's inequality as: $|\langle 2\theta(i, -i), x \rangle| \leq 2\omega \Rightarrow -z\langle 2\theta(i, -i), x \rangle \leq 2\omega$.

Using this in Proposition 4, we have that:

$$\mathrm{Var}[\langle v, z_{-i} \rangle] \geq C\frac{\exp(-8\omega)\|v\|_2^2}{\omega} \tag{14}$$

where $C$ is a universal constant.

Combining (13) and (14), we obtain the statement of the lemma. $\qquad\square$

### B.2.1  Auxiliary Lemmata

**Lemma 6.** *If $w \in \mathbb{R}^{p-1}$ such that $\|w\|_1 \leq 2\omega$, then for $x, y \in \{-1, +1\}^{p-1} \times \{-1, +1\}$:*

$$\sigma(y\langle w, x \rangle)(1 - \sigma(y\langle w, x \rangle)) = \frac{\exp(-y\langle w, x \rangle)}{(1 + \exp(-y\langle w, x \rangle))^2} \geq \frac{\exp(-|y\langle w, x \rangle|)}{4} \geq \frac{\exp(-2\omega)}{4} \tag{15}$$

*Proof.* Consider $f(a) = \sigma(a)(1 - \sigma(a)) = \frac{\exp(-a)}{(1+\exp(-a))^2} = \frac{\exp(a)}{(1+\exp(a))^2}$. Now for $a > 0$:

$$e^{-a} < 1 \Leftrightarrow e^{-a} + 1 < 2 \Leftrightarrow (e^{-a} + 1)^2 < 4 \Leftrightarrow \frac{\exp(-a)}{(1 + \exp(-a))^2} \geq \frac{\exp(-a)}{4}$$

For $a < 0$:

$$e^a < 1 \Leftrightarrow e^a + 1 < 2 \Leftrightarrow (e^a + 1)^2 < 4 \Leftrightarrow \frac{\exp(a)}{(1 + \exp(a))^2} \geq \frac{\exp(a)}{4}$$

Therefore:

$$f(a) \geq \frac{\exp(-|a|)}{4}$$

By Hölder's inequality, $|y\langle w, x \rangle| \leq \|w\|_1 \|x\|_\infty \leq 2\omega$. This implies that

$$\sigma(y\langle w, x \rangle)(1 - \sigma(y\langle w, x \rangle)) = f(y\langle w, x \rangle) \geq \frac{\exp(-|y\langle w, x \rangle|)}{4} \geq \frac{\exp(-2\omega)}{4}$$

$\qquad\square$

# C  Proofs of Propositions in Section 2

In this section, we present the proofs for Theorem 1 and Lemma 1.

## C.1  Proof of Theorem 1

Here, we derive bounds on the modulus of continuity defined in (4) with the loss function given by the $\ell_{2,\infty}$ norm of the parameters.

*Proof Sketch.* We begin by giving a brief proof outline. $\mathbb{P}_{\theta^{(1)}}$ and $\mathbb{P}_{\theta^{(2)}}$ are two Ising models in the high temperature regime (3) with constant $\alpha$, and additionally satisfy $d_{\mathrm{TV}}(\mathbb{P}_{\theta^{(1)}}, \mathbb{P}_{\theta^{(2)}}) \leq \epsilon$. Consider $\mathcal{L}_{\theta^{(1)},i}$ to be the population negative conditional log-likelihood for the $i^{th}$ node with respect to $\mathbb{P}_{\theta^{(1)}}$ defined earlier. We earlier noted that $\nabla \mathcal{L}_{\theta^{(1)},i}(2\theta^{(1)}(i)) = 0$ by the maximum likelihood principle.

In Lemma 7, we show that under these conditions, the gradient $\nabla \mathcal{L}_{\theta^{(1)},i}(2\theta^{(2)}(i))$ satisfies $\|\nabla \mathcal{L}_{\theta^{(1)},i}(2\theta^{(2)}(i))\|_2 \leq \sqrt{C(\alpha)}\epsilon\sqrt{\log(1/\epsilon)}$, where $C(\alpha)$ is a universal constant only depending on $\alpha$. With this intermediate result, we complete the proof of the theorem as follows. Considering the Taylor series expansion of $\mathcal{L}_{\theta^{(1)},i}$ around $2\theta^{(2)}(i)$, we get

$$
\begin{aligned}
\mathcal{L}_{\theta^{(1)},i}(2\theta^{(1)}(i)) &= \mathcal{L}_{\theta^{(1)},i}(2\theta^{(2)}(i)) + \left\langle \nabla \mathcal{L}_{\theta^{(1)}(i)}(2\theta^{(2)}(i)), \Delta_i \right\rangle + \frac{1}{2}\Delta_i^T \nabla^2 \mathcal{L}_{\theta^{(1)},i}(\widetilde{w})\Delta_i \\
&\overset{(i)}{\geq} \mathcal{L}_{\theta^{(1)},i}(2\theta^{(2)}(i)) + \left\langle \nabla \mathcal{L}_{\theta^{(1)}(i)}(2\theta^{(2)}(i)), \Delta_i \right\rangle + \frac{C}{2}\frac{\exp(-O(\omega))}{\omega}\|\Delta_i\|_2^2 \\
&\overset{(ii)}{\geq} \mathcal{L}_{\theta^{(1)},i}(2\theta^{(2)}(i)) + \left\langle \nabla \mathcal{L}_{\theta^{(1)}(i)}(2\theta^{(2)}(i)), \Delta_i \right\rangle + C'\frac{\exp(-c(1-\alpha))}{1-\alpha}\|\Delta_i\|_2^2,
\end{aligned}
$$

where $\widetilde{w}$ lies between $2\theta^{(2)}(i)$ and $2\theta^{(1)}(i)$, and $\Delta_i = 2\theta^{(1)}(i) - 2\theta^{(2)}(i)$. In step $(i)$, we have used the result in Proposition 5 and in step $(ii)$ we use the fact that $\omega \leq 1 - \alpha$.

We also know by the maximum likelihood principle that $\mathcal{L}_{\theta^{(1)},i}(2\theta^{(1)}(i)) \leq \mathcal{L}_{\theta^{(1)},i}(2\theta^{(2)}(i))$, and substituting this in the inequality above yields

$$
C'\frac{\exp(-c(1-\alpha))}{1-\alpha}\|\Delta_i\|_2^2 \leq -\left\langle \nabla \mathcal{L}_{\theta^{(1)}(i)}(2\theta^{(2)}(i)), \Delta_i \right\rangle \leq \left| \left\langle \nabla \mathcal{L}_{\theta^{(1)}(i)}(2\theta^{(2)}(i)), \Delta_i \right\rangle \right|.
$$

Finally, we bound the right hand side using the Cauchy-Schwarz inequality and the result from Lemma 7 to get

$$
\left| \left\langle \nabla \mathcal{L}_{\theta^{(1)}(i)}(2\theta^{(2)}(i)), \Delta_i \right\rangle \right| \leq \|\nabla \mathcal{L}_{\theta^{(1)},i}(2\theta^{(2)}(i))\|_2 \|\Delta_i\|_2 \leq \sqrt{C(\alpha)}\epsilon\sqrt{\log(1/\epsilon)}\|\Delta_i\|_2,
$$

and substituting this in the quadratic bound above gives

$$
\|\Delta_i\|_2 \leq C_1(\alpha)\epsilon\sqrt{\log(1/\epsilon)}, \qquad C_1(\alpha) = \frac{1}{C'}(1-\alpha)\exp(c(1-\alpha))\sqrt{C(\alpha)}.
$$

$\square$

We now state Lemma 7 and prove it below.

**Lemma 7.** *Let $\mathbb{P}_{\theta^{(1)}}$ and $\mathbb{P}_{\theta^{(2)}}$ be two Ising models in the high temperature regime (3) with constant $\alpha$ that satisfies $d_{\mathrm{TV}}(\mathbb{P}_{\theta^{(1)}}, \mathbb{P}_{\theta^{(2)}}) \leq \epsilon$. Then, there exists a universal constant $C(\alpha)$ that only depends on $\alpha$ such that*

$$
\|\nabla \mathcal{L}_{\theta^{(1)},i}(2\theta^{(2)}(i))\|_2 \leq \sqrt{C(\alpha)}\epsilon\sqrt{\log(1/\epsilon)} \qquad \text{for all } i \in [p]
$$

*Proof.* Recall that $\mathcal{L}_{\theta^{(1)},i}(w) = \mathbb{E}_{z \sim \mathbb{P}_{\theta^{(1)}}}[\ell_i(w;z)]$. By the maximum likelihood principle, we know that

$$
\nabla \mathcal{L}_{\theta^{(1)},i}(2\theta^{(1)}(i)) = \mathbf{0} \qquad \nabla \mathcal{L}_{\theta^{(2)},i}(2\theta^{(2)}(i)) = \mathbf{0}
$$

Since $d_{\mathrm{TV}}(\mathbb{P}_{\theta^{(1)}}, \mathbb{P}_{\theta^{(2)}}) \leq \epsilon$, there exists an $\epsilon$-coupling $\mathcal{C}$ between $\mathbb{P}_{\theta^{(1)}}$ and $\mathbb{P}_{\theta^{(2)}}$. In particular, $\mathcal{C}$ is a joint distribution over $z_1, z_2$ such that the respective marginals are $z_1 \sim \mathbb{P}_{\theta^{(1)}}$ and $z_2 \sim \mathbb{P}_{\theta^{(2)}}$, and $\mathbb{E}_{z_1, z_2 \sim \mathcal{C}}[\mathbb{I}\{z_1 \neq z_2\}] \leq \epsilon$.

The rest of the proof begins by making the observation that $\nabla \mathcal{L}_{\theta^{(1)}, i}(2\theta^{(2)}(i)) = \mathbb{E}_{z_1, z_2 \sim \mathcal{C}}[\nabla \ell_i(2\theta^{(2)}(i); z_1)]$. By introducing indicator random variables for the cases when $z_1$ and $z_2$ are equal or not, we have

$$
\begin{aligned}
\nabla \mathcal{L}_{\theta^{(1)}, i}(2\theta^{(2)}(i)) &= \mathbb{E}_{z_1, z_2 \sim \mathcal{C}}[\nabla \ell_i(2\theta^{(2)}(i); z_1)\mathbb{I}\{z_1 \neq z_2\}] + \mathbb{E}_{z_1, z_2 \sim \mathcal{C}}[\nabla \ell_i(2\theta^{(2)}(i); z_1)\mathbb{I}\{z_1 = z_2\}] \\
&= \mathbb{E}_{z_1, z_2 \sim \mathcal{C}}[\nabla \ell_i(2\theta^{(2)}(i); z_1)\mathbb{I}\{z_1 \neq z_2\}] + \mathbb{E}_{z_1, z_2 \sim \mathcal{C}}[\nabla \ell_i(2\theta^{(2)}(i); z_2)\mathbb{I}\{z_1 = z_2\}] \\
&\overset{(a)}{=} \mathbb{E}_{z_1, z_2 \sim \mathcal{C}}[\nabla \ell_i(2\theta^{(2)}(i); z_1)\mathbb{I}\{z_1 \neq z_2\}] - \mathbb{E}_{z_1, z_2 \sim \mathcal{C}}[\nabla \ell_i(2\theta^{(2)}(i); z_2)\mathbb{I}\{z_1 \neq z_2\}],
\end{aligned}
$$

where step $(a)$ follows from the stationarity of $2\theta^{(2)}(i)$ under $\mathbb{P}_{\theta^{(2)}}$ like so.

$$
\begin{aligned}
\mathbf{0} &= \nabla \mathcal{L}_{\theta^{(2)}, i}(2\theta^{(2)}(i)) \\
&= \mathbb{E}_{z_1, z_2 \sim \mathcal{C}}[\nabla \ell_i(2\theta^{(2)}(i); z_2)] \\
&= \mathbb{E}_{z_1, z_2 \sim \mathcal{C}}[\nabla \ell_i(2\theta^{(2)}(i); z_2)\mathbb{I}\{z_1 = z_2\}] + \mathbb{E}_{z_1, z_2 \sim \mathcal{C}}[\nabla \ell_i(2\theta^{(2)}(i); z_2)\mathbb{I}\{z_1 \neq z_2\}].
\end{aligned}
$$

Therefore, for any vector $v \in \mathcal{S}^{p-2}$, we have that

$$
\begin{aligned}
\left| \left\langle v, \nabla \mathcal{L}_{\theta^{(1)}, i}(2\theta^{(2)}(i)) \right\rangle \right| &= \left| \mathbb{E}_{z_1, z_2 \sim \mathcal{C}}[\left\langle v, \nabla \ell_i(2\theta^{(2)}(i); z_1) \right\rangle \mathbb{I}\{z_1 \neq z_2\}] \right. \\
&\qquad \left. -\mathbb{E}_{z_1, z_2 \sim \mathcal{C}}[\left\langle v, \nabla \ell_i(2\theta^{(2)}(i); z_2) \right\rangle \mathbb{I}\{z_1 \neq z_2\}] \right| \\
&\leq \underbrace{\left| \mathbb{E}_{z_1, z_2 \sim \mathcal{C}}[\left\langle v, \nabla \ell_i(2\theta^{(2)}(i); z_1) \right\rangle \mathbb{I}\{z_1 \neq z_2\}] \right|}_{T_1} \\
&\quad + \underbrace{\left| \mathbb{E}_{z_1, z_2 \sim \mathcal{C}}[\left\langle v, \nabla \ell_i(2\theta^{(2)}(i); z_2) \right\rangle \mathbb{I}\{z_1 \neq z_2\}] \right|}_{T_2}.
\end{aligned}
$$

**Bounding $T_2$:** Note that $\nabla \ell_i(w; z_1) = (\sigma(\langle w, z_1(-i)\rangle z_1(i)) - 1)z_1(-i)z_1(i)$. Since $z_1 \sim \{-1, +1\}^p$, we have that $|(\sigma(\langle w, z_1(-i)\rangle z_1(i)) - 1)z_1(i)| < 1$, and hence we get $|\langle v, \nabla \ell_i(w; z_1)| < |\langle v, z_1(-i)\rangle|$.

This in turn implies

$$
\Pr(|\langle v, \nabla \ell_i(w; z_1)\rangle| > t) \leq \Pr(|\langle v, z_1(-i)\rangle| > t) \overset{(b)}{\leq} 2\exp\left(-\frac{t^2}{C(\alpha)}\right)
$$

where step $(b)$ follows from the sub-Gaussianity of random variables distributed with respect to an Ising model in the high temperature regime (Proposition 3). Using standard tail bounds (see [49, Chapter 2]), we obtain that $\mathbb{E}[\exp(\lambda(\langle v, \nabla \ell_i(w; z_1)\rangle))] \leq \exp\left(\frac{C\lambda^2 C(\alpha)}{2}\right)$. To finally bound $T_2$, we use the following result from [39].

**Proposition 8** ([39, Lemma 2.3]). *Let $Z$ be a random variable such that $\mathbb{E}[\exp(\lambda Z)] \leq e^{\frac{\lambda^2 \sigma^2}{2}}$. For any measurable event $A$, we have*

$$
|\mathbb{E}[Z \cdot \mathbb{I}\{A\}]| \leq \sigma P(A)\sqrt{\log(1/P(A))}.
$$

In $T_2$, the event $A$ is $z_1 \neq z_2$ and this occurs with probability less than $\epsilon$. Hence, we get $T_2 \leq C\sqrt{C(\alpha)}\epsilon\sqrt{\log(1/\epsilon)}$.

**Bounding** $T_1$**:** This can be bounded in an analogous manner as $T_2$, thus yielding $T_1 \leq C\sqrt{C(\alpha)}\epsilon\sqrt{\log(2/\epsilon)}$.

Plugging these bounds above, we get

$$\|\nabla \mathcal{L}_{\theta^{(1)},i}(2\theta^{(2)}(i))\|_2 \leq C\sqrt{C(\alpha)}\epsilon\sqrt{\log(1/\epsilon)},$$

which proves the statement of the lemma. $\qquad\square$

### C.2 Proof of Lemma 1

*Proof.* Consider two Ising models with $p$ vertices. For the first Ising model, consider one edge with parameter $2\epsilon$. The second Ising model has no edges.

Via a simple calculation, the TV distance between these Ising models can be computed to be $\frac{1}{2}\tanh(2\epsilon) \leq \epsilon$. Consequently, the $\ell_{2,\infty}$ norm of the difference in parameters is $\epsilon$, and this proves the lower bound. $\qquad\square$

# D Proofs of Propositions in Section 3

## D.1 A general result for estimators based on Yatracos classes

Here, we present a result for estimators of the form

$$\mathbb{P}_{\text{est}} = \underset{\mathbb{P} \in \mathcal{P}}{\operatorname{argmin}} \, \underset{A \in \mathcal{A}}{\sup} \left| \mathbb{P}(A) - \widehat{\mathbb{P}}_{n,\epsilon}(A) \right|, \tag{16}$$

where $\widehat{\mathbb{P}}_{n,\epsilon}$ the empirical distribution of $n$ samples from the mixture model $\mathbb{P}_\epsilon$ defined in (1) and $\mathcal{P}$ is the class of all distributions. Recall that $\mathcal{A}$ is defined as

$$\mathcal{A} = \{A(\mathbb{P}_1, \mathbb{P}_2) : \mathbb{P}_1, \mathbb{P}_2 \in \mathcal{P}\}, \text{ and } A(\mathbb{P}_1, \mathbb{P}_2) = \{x : \mathbb{P}_1(x) > \mathbb{P}_2(x)\}$$

The result in formally stated in Proposition 2.

**Proposition 9.** *Given $n$ samples from the mixture model $\mathbb{P}_\epsilon = (1 - \epsilon)\mathbb{P}^\star + \epsilon Q$, the estimator $\mathbb{P}_{\text{est}}$ defined in (16) satisfies*

$$d_{\text{TV}}(\mathbb{P}_{\text{est}}, \mathbb{P}^\star) \leq 2\epsilon + 2 \sup_{A \in \mathcal{A}} \left| \sum_{x \in A} \widehat{\mathbb{P}}_{n,\epsilon}(x) - \sum_{x \in A} \mathbb{P}_\epsilon(x) \right|$$

*Proof.* We begin by using $2d_{\text{TV}}(\mathbb{P}_{\text{est}}, \mathbb{P}^\star) = \sum_{x \in \mathcal{X}} |\mathbb{P}_{\text{est}}(x) - \mathbb{P}^\star(x)|$. Consider the sets $B = \{x : \mathbb{P}_{\text{est}}(x) > \mathbb{P}^\star(x)\}$ and $C = \{x : \mathbb{P}_{\text{est}}(x) \leq \mathbb{P}^\star(x)\}$.

This gives us:

$$
\begin{aligned}
\sum_{x \in \mathcal{X}} |\mathbb{P}_{\text{est}}(x) - \mathbb{P}^\star(x)| &= 2 \max_{A \in \{B,C\}} \left| \sum_{x \in A} \mathbb{P}_{\text{est}}(x) - \mathbb{P}^\star(x) \right| \\
&\leq 2 \sup_{A \in \mathcal{A}} \left| \sum_{x \in A} \mathbb{P}_{\text{est}}(x) - \sum_{x \in A} \mathbb{P}^\star(x) \right| \\
&= 2 \sup_{A \in \mathcal{A}} \left| \sum_{x \in A} \mathbb{P}_{\text{est}}(x) - \sum_{x \in A} \widehat{\mathbb{P}}_{n,\epsilon}(x) + \sum_{x \in A} \widehat{\mathbb{P}}_{n,\epsilon}(x) - \sum_{x \in A} \mathbb{P}^\star(x) \right| \\
&\leq 2 \sup_{A \in \mathcal{A}} \left| \sum_{x \in A} \mathbb{P}_{\text{est}}(x) - \sum_{x \in A} \widehat{\mathbb{P}}_{n,\epsilon}(x) \right| + 2 \sup_{A \in \mathcal{A}} \left| \sum_{x \in A} \widehat{\mathbb{P}}_{n,\epsilon}(x) - \sum_{x \in A} \mathbb{P}^\star(x) \right| \\
&\overset{(i)}{\leq} 4 \sup_{A \in \mathcal{A}} \left| \sum_{x \in A} \widehat{\mathbb{P}}_{n,\epsilon}(x) - \sum_{x \in A} \mathbb{P}^\star(x) \right| \\
&= 4 \sup_{A \in \mathcal{A}} \left| \sum_{x \in A} \widehat{\mathbb{P}}_{n,\epsilon}(x) - \sum_{x \in A} \mathbb{P}_\epsilon(x) + \sum_{x \in A} \mathbb{P}_\epsilon(x) - \sum_{x \in A} \mathbb{P}^\star(x) \right| \\
&\leq 4 \sup_{A \in \mathcal{A}} \left| \sum_{x \in A} \widehat{\mathbb{P}}_{n,\epsilon}(x) - \sum_{x \in A} \mathbb{P}_\epsilon(x) \right| + 4 \sup_{A \in \mathcal{A}} \left| \sum_{x \in A} \mathbb{P}_\epsilon(x) - \sum_{x \in A} \mathbb{P}^\star(x) \right| \\
&= 4 \sup_{A \in \mathcal{A}} \left| \sum_{x \in A} \widehat{\mathbb{P}}_{n,\epsilon}(x) - \sum_{x \in A} \mathbb{P}_\epsilon(x) \right| + 4 d_{\text{TV}}(\mathbb{P}_\epsilon, \mathbb{P}^\star) \\
&\overset{(ii)}{\leq} 4 \sup_{A \in \mathcal{A}} \left| \sum_{x \in A} \widehat{\mathbb{P}}_{n,\epsilon}(x) - \sum_{x \in A} \mathbb{P}_\epsilon(x) \right| + 4\epsilon,
\end{aligned}
$$

where in step $(i)$ we have used the optimality of $\mathbb{P}_{\text{est}}$ and in step $(ii)$ we have used the fact that $d_{\text{TV}}(\mathbb{P}_\epsilon, \mathbb{P}^\star) \leq \epsilon$ and this completes the proof. $\qquad \square$

## D.2 Proof of Lemma 2

With the general result for estimators based on Yatracos classes, we state the proof of Lemma 2.

*Proof.* For the estimator in (7), the class of distributions is $\mathcal{G}_{p,k}$. Via Proposition 9, we have that:

$$d_{\text{TV}}(\mathbb{P}_{\widehat{\theta}}, \mathbb{P}_{\theta^\star}) \leq 2\epsilon + 2 \underbrace{\sup_{A \in \mathcal{A}} \left| \sum_{x \in A} \widehat{\mathbb{P}}_{n,\epsilon}(x) - \sum_{x \in A} \mathbb{P}_\epsilon(x) \right|}_{T_1}$$

Note that distributions in $\mathcal{G}_{p,k}$ are Ising model distributions and are parameterized. Thus, we can alternatively identify the sets $A(\mathbb{P}_1, \mathbb{P}_2)$ via the parameters of Ising model distributions as $A(\theta^{(1)}, \theta^{(2)})$.

**Bounding $T_1$:** The set $A(\theta^{(1)}, \theta^{(2)})$ is equivalent to

$$A(\theta^{(1)}, \theta^{(2)}) = \{x : \log \mathbb{P}_{\theta^{(1)}}(x) > \log \mathbb{P}_{\theta^{(2)}}(x)\}$$

Recalling the definitions of $\mathbb{P}_{\theta^{(1)}}$ and $\mathbb{P}_{\theta^{(2)}}$, and flattening the parameters to $\mathbb{R}^{\binom{p}{2}}$, we have:

$$A(\theta^{(1)}, \theta^{(2)}) = \left\{ y : \left\langle \theta^{(1)}_{\text{flat}} - \theta^{(2)}_{\text{flat}}, y \right\rangle + \log(Z(\theta^{(2)})) - \log(Z(\theta^{(1)})) > 0 \right\} = \{y : \langle w, \widetilde{y} \rangle > 0\}$$

where $w = [\theta^{(1)}_{\text{flat}} - \theta^{(2)}_{\text{flat}}, \log(Z(\theta^{(2)})) - \log(Z(\theta^{(1)}))]$ and $\tilde{y} = [y, 1]$. $Z(\theta)$ is the normalization constant of the probability mass function of an Ising model $\mathbb{P}_\theta$ and $y \in \mathbb{R}^{\binom{p}{2}}$ is a vector of sufficient statistics. Since $\theta^{(1)}, \theta^{(2)} \in \mathcal{G}_{p,k}$, both $\theta^{(1)}_{\text{flat}}$ and $\theta^{(2)}_{\text{flat}}$ can have at most $k$ entries. Consequently, the vector $w$ can have at most $2k + 1$ non-zero entries. Hence, $\mathcal{A}$ can be viewed as a collection of sets:

$$\mathcal{A} = \{\mathbb{I}\{\langle w, y \rangle > 0\} : w \in \mathbb{R}^{\binom{p}{2}}, ||w||_0 \leq 2k + 1\}$$

The following proposition bounds the VC-dimension of sparse linear classifiers:

**Proposition 10** ([1, Corollary 1])**.** *Consider the class of linear predictors, defined by the set $S_s = \{v : ||v||_0 \leq s, v \in \mathbb{R}^m\}$ i.e. the set of $s$-sparse vectors. The VC-dimension of this class is upper bounded as: $O(s \log(\text{em}/s))$.*

Therefore, from the above proposition, we have that the VC-dimension of $\mathcal{A}$ is bounded from above by $\mathcal{O}(2k + 1) \log(\text{ep}^2/4k+2)$ which is $\mathcal{O}(k \log(\text{ep}/k))$. Hence, by a concentration of measure argument, we have that with probability at least $1 - \delta$:

$$T_1 \lesssim \sqrt{\frac{k \log(\text{ep}/k)}{n}} + \sqrt{\frac{\log(1/\delta)}{n}}.$$

Finally, we obtain

$$d_{\text{TV}}(\mathbb{P}_{\widehat{\theta}}, \mathbb{P}_{\theta^\star}) \leq 2\epsilon + \mathcal{O}\left( \sqrt{\frac{k \log(\text{ep}/k)}{n}} + \sqrt{\frac{\log(1/\delta)}{n}} \right),$$

and this recovers the statement of the lemma. □

# E  Proof of Propositions in Section 4

## E.1  Proof of Theorem 2

*Proof Sketch.* We give an outline of the proof of the theorem. $\mathbb{P}_{\theta^\star}$ is an Ising model in the high temperature regime with constant $\alpha$. Recall the proposed estimator:

$$\widehat{\theta}(i) = \underset{w \in \mathcal{N}_d^\gamma(\mathcal{S}^{p-2})}{\operatorname{argmin}} \; \underset{u \in \mathcal{N}_{2d}^{1/2}(\mathcal{S}^{p-2})}{\sup} \left| \mathsf{1DMean}\left( \{u^T \nabla \ell_i(w; x^{(j)})\}_{j=1}^n \right) \right|. \tag{17}$$

Proposition 5 states that the negative conditional log-likelihood $\mathcal{L}_{\theta^\star,i}$ is $C_2(\alpha)$-strongly convex, where $C_2(\alpha)$ is a universal constant only depending on $\alpha$. Therefore, by the monotonicity of the gradient of strongly-convex function, we bound the parameter error $\|\widehat{\theta}(i) - \theta^\star(i)\|_2$ as

$$\|\widehat{\theta}(i) - \theta^\star(i)\|_2^2 \leq \frac{1}{C_2(\alpha)} \left\langle \nabla \mathcal{L}_{\theta^\star,i}(\widehat{\theta}(i)) - \nabla \mathcal{L}_{\theta^\star,i}(\theta^\star(i)), \widehat{\theta}(i) - \theta^\star(i) \right\rangle.$$

Next, note that

$$
\begin{aligned}
\|\widehat{\theta}(i) - \theta^\star(i)\|_2 &\leq \frac{1}{C_2(\alpha)} \frac{\left\langle \nabla \mathcal{L}_{\theta^\star,i}(\widehat{\theta}(i)) - \nabla \mathcal{L}_{\theta^\star,i}(\theta^\star(i)), \widehat{\theta}(i) - \theta^\star(i) \right\rangle}{\|\widehat{\theta}(i) - \theta^\star(i)\|_2} \\
&\overset{(i)}{\leq} \frac{1}{C_2(\alpha)} \underset{u \in \mathcal{N}_{2d}(\mathcal{S}^{p-2})}{\sup} \left| \left\langle u, \nabla \mathcal{L}_{\theta^\star,i}(\widehat{\theta}(i)) \right\rangle \right| \\
&\overset{(ii)}{\leq} \frac{2}{C_2(\alpha)} \underset{u \in \mathcal{N}_{2d}^{1/2}(\mathcal{S}^{p-2})}{\sup} \left| \left\langle u, \nabla \mathcal{L}_{\theta^\star,i}(\widehat{\theta}(i)) \right\rangle \right|,
\end{aligned}
$$

where in step $(i)$ we have used the facts that 1) $\frac{\widehat{\theta}(i) - \theta^\star(i)}{\|\widehat{\theta}(i) - \theta^\star(i)\|_2}$ is a unit vector with at most $2d$ non-zero elements, and 2) $\nabla \mathcal{L}_{\theta^\star,i}(\theta^\star(i)) = \mathbf{0}$ by the maximum likelihood principle, and in step $(ii)$ we have constructed a $1/2$-cover of the set $\mathcal{N}_{2d}^{1/2}(\mathcal{S}^{p-2})$.

We further analyze the right hand side by splitting it into two different terms as follows.

$$
\begin{aligned}
\underset{u \in \mathcal{N}_{2d}^{1/2}(\mathcal{S}^{p-2})}{\sup} & \left| \left\langle u, \mathcal{L}_{\theta^\star,i}(\widehat{\theta}(i)) \right\rangle \right| \leq \\
& \underbrace{\underset{u \in \mathcal{N}_{2d}^{1/2}(\mathcal{S}^{p-2})}{\sup} \left| \left\langle u, \mathcal{L}_{\theta^\star,i}(\widehat{\theta}(i)) \right\rangle - \mathsf{1DMean}\left( \{u^T \nabla \ell_i(\widehat{\theta}(i), x^{(j)})\}_{j=1}^n \right) \right|}_{T_1} \\
& + \underbrace{\underset{u \in \mathcal{N}_{2d}^{1/2}(\mathcal{S}^{p-2})}{\sup} \left| \mathsf{1DMean}\left( \{u^T \nabla \ell_i(\widehat{\theta}(i), x^{(j)})\}_{j=1}^n \right) \right|}_{T_2}.
\end{aligned}
$$

In Lemmas 11 and 12, considering $\gamma = \max\left\{ \frac{\epsilon}{p}, \frac{\log(1/\delta)}{np} \right\}$, and for sufficiently large $n$ (10), we bound $T_1$ and $T_2$ as $T_1 \leq \sqrt{C(\alpha)} \left\{ \epsilon \sqrt{\log\left(\frac{1}{\epsilon}\right)} + \sqrt{\frac{d \log(p)}{n}} + \sqrt{\frac{d}{n} \log\left(\frac{3ep}{d\gamma}\right)} \right\}$, and in Lemma 12, we bound $T_2$ as $T_2 \leq \sqrt{C(\alpha)} \left\{ \epsilon \sqrt{\log\left(\frac{1}{\epsilon}\right)} + \sqrt{\frac{d \log(p)}{n}} + \sqrt{\frac{d}{n} \log\left(\frac{3ep}{d\gamma}\right)} \right\} + \max\left( \epsilon, \frac{\log(1/\delta)}{n} \right)$ respectively.

Plugging these bound into the previous right hand side, we obtain

$$\|\widehat{\theta}(i)-\theta^\star(i)\|_2 \lesssim \sqrt{C(\alpha)}\left\{\epsilon\sqrt{\log\left(\frac{1}{\epsilon}\right)} + \sqrt{\frac{d\log(p)}{n}} + \sqrt{\frac{d}{n}\log\left(\frac{3ep}{d\gamma}\right)}\right\} + \max\left(\epsilon, \frac{\log(1/\delta)}{n}\right),$$

and this recovers the statement of the theorem. $\qquad\square$

We state Lemmas 11 and 12 and prove them below.

**Lemma 11.** *Consider samples $\{x^{(j)}\}_{j=1}^n$ from the mixture model $\mathbb{P}_\epsilon = (1-\epsilon)\mathbb{P}_{\theta^\star} + \epsilon Q$, where $\mathbb{P}_{\theta^\star}$ is an Ising model over $p$ nodes in the high temperature regime (3) with constant $\alpha$ and with maximum vertex degree $d$. Suppose $n$, confidence $\delta$ and contamination level $\epsilon$ satisfy (10). Then,* 1DMean *satisfies*

$$\sup_{w\in\mathcal{N}_d^\gamma(\mathcal{S}^{p-2})} \sup_{u\in\mathcal{N}_d^{1/2}(\mathcal{S}^{p-2})} \left|\langle u, \nabla\mathcal{L}_{\theta^\star,i}(w)\rangle - \text{1DMean}\left(\{u^T\nabla\ell_i(w; x^{(j)})\}_{j=1}^n\right)\right|$$

$$\leq \sqrt{C(\alpha)}\left\{\epsilon\sqrt{\log\left(\frac{1}{\epsilon}\right)} + \sqrt{\frac{d\log(p)}{n}} + \sqrt{\frac{d}{n}\log\left(\frac{3ep}{d\gamma}\right)}\right\}.$$

*Proof.* Let $z \sim \mathbb{P}_{\theta^\star}$. In the proof of Lemma 7, we showed that

$$\Pr(|\langle u, \nabla\ell_i(w;z)\rangle|) \leq 2\exp\left(-\frac{t^2}{C(\alpha)}\right)$$

holds due to the form of the gradient and the sub-Gaussianity of the Ising model distribution. This implies that the gradients of $\ell_i$ due to non-outlier samples are sub-Gaussian. This allows us to leverage techniques from [41] to produce a guarantee for the 1DMean algorithm when the true distribution is sub-Gaussian in Lemma 13. This states that

$$\left|\langle u, \nabla\mathcal{L}_{\theta^\star,i}(w)\rangle - \text{1DMean}\left(\{u^T\nabla\ell_i(w; x^{(j)})\}_{j=1}^n\right)\right| \lesssim \epsilon\sqrt{C(\alpha)\log\left(\frac{1}{\epsilon}\right)} + \sqrt{\frac{C(\alpha)}{n}\log\left(\frac{1}{\delta}\right)},$$

where $w \in \mathcal{N}_d^\gamma(\mathcal{S}^{p-2})$ and $u \in \mathcal{N}_d^{1/2}(\mathcal{S}^{p-2})$.

Finally, to convert the point-wise bound to a uniform bound, we perform a union bound over all the elements in $\mathcal{N}_d^\gamma(\mathcal{S}^{p-2})$ and $\mathcal{N}_d^{1/2}(\mathcal{S}^{p-2})$, and use the fact that the number of elements in the cover can be bounded as $|\mathcal{N}_k^\gamma(\mathcal{S}^{p-2})| \leq \left(\frac{3ep}{k\gamma}\right)^k$ to recover the statement of the result. $\qquad\square$

**Lemma 12.** *Given samples $\{x^{(j)}\}_{j=1}^n$ from the mixture model $\mathbb{P}_\epsilon = (1-\epsilon)\mathbb{P}_{\theta^\star} + \epsilon\mathbb{Q}$, where $\mathbb{P}_{\theta^\star}$ is an Ising model over $p$ nodes in the high temperature regime (3) with constant $\alpha$, there exists a constant $C(\alpha)$ that only depends on $\alpha$ such that:*

$$\sup_{u\in\mathcal{N}_{2d}^{1/2}(\mathcal{S}^{p-2})} \left|\text{1DMean}\left(\{u^T\nabla\ell_i(\widehat{\theta}(i); x^{(j)}\}_{j=1}^n\right)\right|$$

$$\leq \sqrt{C(\alpha)}\left\{\epsilon\sqrt{\log\left(\frac{1}{\epsilon}\right)} + \sqrt{\frac{d\log(p)}{n}} + \sqrt{\frac{d}{n}\log\left(\frac{3ep}{d\gamma}\right)}\right\} + \max\left(\epsilon, \frac{\log(1/\delta)}{n}\right)$$

*where $\widehat{\theta}(i)$ is as defined in (9) with $\gamma = \max\left\{\frac{\epsilon}{p}, \frac{\log(1/\delta)}{p}\right\}$.*

*Proof.* First, define $C_\gamma(\theta^\star(i))$ as the element closest to $\theta^\star(i)$ in the set $\mathcal{N}_d^\gamma(\mathcal{S}^{p-2})$.

We begin the proof by recognizing that

$$\sup_{u \in \mathcal{N}_{2d}^{1/2}(\mathcal{S}^{p-2})} \left| \text{1DMean}\left(\{u^T \nabla \ell_i(\widehat{\theta}(i); x^{(j)})\}_{j=1}^n\right) \right|$$

$$\overset{(i)}{\leq} \sup_{u \in \mathcal{N}_{2d}^{1/2}(\mathcal{S}^{p-2})} \left| \text{1DMean}\left(\{u^T \nabla \ell_i(C_\gamma(\theta^\star(i)); x^{(j)})\}_{j=1}^n\right) \right|$$

$$\overset{(ii)}{\leq} \underbrace{\sup_{u \in \mathcal{N}_{2d}^{1/2}(\mathcal{S}^{p-2})} \left| \text{1DMean}\left(\{u^T \nabla \ell_i(C_\gamma(\theta^\star(i)); x^{(j)})\}_{j=1}^n\right) - \langle u, \nabla \mathcal{L}_{\theta^\star, i}(C_\gamma(\theta^\star(i)))\rangle \right|}_{T_{2,1}}$$

$$+ \underbrace{\sup_{u \in \mathcal{N}_{2d}^{1/2}(\mathcal{S}^{p-2})} |\langle u, \nabla \mathcal{L}_{\theta^\star, i}(C_\gamma(\theta^\star(i)))\rangle|}_{T_{2,2}}$$

where Step $(i)$ uses the optimality of $\widehat{\theta}(i)$ and Step $(ii)$ performs splitting by addition and subtraction as mentioned earlier.

**Bounding $T_{2,1}$:**  $T_{2,1}$ can be bounded using Lemma 11, since it holds for any $w \in \mathcal{N}_d^\gamma(\mathcal{S}^{p-2})$ and $C_\gamma(\theta^\star(i)) \in \mathcal{N}_d^\gamma(\mathcal{S}^{p-2}$ by definition. Therefore, we get

$$\sup_{u \in \mathcal{N}_{2d}^{1/2}(\mathcal{S}^{p-2})} \left| \text{1DMean}\left(\{u^T \nabla \ell_i(C_\gamma(\theta^\star(i)); x^{(j)})\}_{j=1}^n\right) - \langle u, \nabla \mathcal{L}_{\theta^\star, i}(C_\gamma(\theta^\star(i)))\rangle \right|$$

$$\leq \sqrt{C(\alpha)} \left\{ \epsilon\sqrt{\log\left(\frac{1}{\epsilon}\right)} + \sqrt{\frac{d \log(p)}{n}} + \sqrt{\frac{d}{n} \log\left(\frac{3ep}{d\gamma}\right)} \right\}.$$

**Bounding $T_{2,2}$:**  $T_{2,2}$ can be bounded as follows:

$$\sup_{u \in \mathcal{N}_{2d}^{1/2}(\mathcal{S}^{p-2})} |\langle u, \nabla \mathcal{L}_{\theta^\star, i}(C_\gamma(\theta^\star(i)))\rangle| \leq \|\nabla \mathcal{L}_{\theta^\star, i}(C_\gamma(\theta^\star(i)))\|_2$$

$$= \|\nabla \mathcal{L}_{\theta^\star, i}(C_\gamma(\theta^\star(i))) - \nabla \mathcal{L}_{\theta^\star, i}(\theta^\star(i))\|_2$$
$$\leq L\|C_\gamma(\theta^\star(i)) - \theta^\star(i)\|_2 \leq L\gamma,$$

where $L$ is the Lipschitz constant of $\mathcal{L}_{\theta^\star, i}$. A simple calculation reveals that:

$$\nabla^2 \mathcal{L}_{\theta^\star, i}(w) = \mathbb{E}_{x \sim P_{\theta^\star}}\left[\sigma(\langle w, x(-i)\rangle x_i)(1 - \sigma(\langle w, x(-i)\rangle x_i))x(-i)x(-i)^T\right]$$
$$\Rightarrow v^T \nabla^2 \mathcal{L}_{\theta^\star, i}(w) v = \mathbb{E}_{x \sim P_{\theta^\star}}\left[\sigma(\langle w, x(-i)\rangle x_i)(1 - \sigma(\langle w, x(-i)\rangle x_i))(\langle v, x(-i)\rangle)^2\right]$$
$$\overset{(i)}{\leq} \frac{1}{4}\mathbb{E}_{x \sim P_{\theta^\star}}[(v^T x_i)^2] \overset{(ii)}{\leq} \frac{p}{4}\|v\|_2^2$$

where in Step $(i)$ we have used the fact that $\sigma(z)(1 - \sigma(z)) \leq \frac{1}{4}$ and in Step $(ii)$ we have used the Cauchy-Schwarz inequality, leading to $L = p$.

With the choice of $\gamma = \max\left\{\frac{\epsilon}{p}, \frac{\log(1/\delta)}{n}\right\}$, we have the final result

$$\sup_{u \in \mathcal{N}_{2d}^{1/2}(\mathcal{S}^{p-2})} \left| \text{1DMean}\left(\{u^T \nabla \ell_i(\widehat{\theta}(i); x^{(j)})\}_{j=1}^n\right) \right|$$

$$\leq \sqrt{C(\alpha)} \left\{ \epsilon\sqrt{\log\left(\frac{1}{\epsilon}\right)} + \sqrt{\frac{d \log(p)}{n}} + \sqrt{\frac{d}{n} \log\left(\frac{3ep}{d\gamma}\right)} \right\} + \max\left(\epsilon, \frac{\log(1/\delta)}{n}\right)$$

and this completes the proof. $\square$

### E.1.1 Auxiliary Results

Here we state and prove Lemma 13, which we use in the proof of Lemma 11.

**Lemma 13** ([41, Lemma 3]). *Suppose $\mathbb{P}^\star$ is a sub-Gaussian distribution with variance proxy $\sigma^2$ and mean $\mu = \mathbb{E}_{x \sim \mathbb{P}^\star}[x]$. Given $n$ samples from the mixture distribution $\mathbb{P}_\epsilon = (1 - \epsilon)\mathbb{P}^\star + \epsilon Q$, Algorithm 1 returns an estimate $\widehat{\theta}_\delta$ that satisfies*

$$|\widehat{\theta}_\delta - \mu| \lesssim \epsilon \sqrt{\sigma^2 \log\left(\frac{1}{\epsilon}\right)} + \sqrt{\sigma^2 \log\left(\frac{1/\delta}{n}\right)}$$

*with probability at least $1 - \delta$.*

*Proof.* The proof mostly follows the proof in [41].

Let $I^\star$ be the interval $\mu \pm \sqrt{\sigma^2 \log\left(\frac{1}{\delta_1}\right)}$. For notational convenience, let $f_n(u, v) = \sqrt{u(1-u)}\sqrt{\frac{\log(1/v)}{n}} + \frac{2}{3}\frac{\log(1/v)}{n}$. Let $\widehat{I} = [a, b]$ be the interval obtained using the first split of the sample set $\mathcal{Z}_1$ *i.e.* the shortest interval containing $n(1 - (\delta_1 + \epsilon + f_n(\epsilon + \delta_1, \delta_3)))$ points of $\mathcal{Z}_1$. In Algorithm 1, we have $\delta_1 = \epsilon$ and $\delta_3 = \delta/4$.

From [41, Claim 5], we have that

$$\text{length}(\widehat{I}) \leq \text{length}(I^\star) \leq 2\sqrt{\sigma^2 \log\left(\frac{1}{\delta_1}\right)}.$$

To bound the error of the estimator, we analyze the quantity

$$\left|\frac{1}{|\widehat{I}|} \sum_{z_i \in \mathcal{Z}_2} z_i \mathbb{I}\left\{z_i \in \widehat{I}\right\} - \mu\right|,$$

where $|\widehat{I}| = \sum_{z_i \in \mathcal{Z}_2} \mathbb{I}\left\{z_i \in \widehat{I}\right\}$.

We do so by casing on whether a sample $z_i$ was sampled from $\mathbb{P}^\star$ or from $Q$, like so.

$$\left|\frac{1}{|\widehat{I}|} \sum_{z_i \in \mathcal{Z}_2} z_i \mathbb{I}\left\{z_i \in \widehat{I}\right\} - \mu\right| = \left|\frac{1}{|\widehat{I}|}\left(\sum_{\substack{z_i \in \mathcal{Z}_2 \\ z_i \sim \mathbb{P}^\star}} z_i \mathbb{I}\left\{z_i \in \widehat{I}\right\} + \sum_{\substack{z_i \in \mathcal{Z}_2 \\ z_i \sim Q}} z_i \mathbb{I}\left\{z_i \in \widehat{I}\right\}\right) - \mu\right|$$

$$\leq \underbrace{\left|\frac{1}{|\widehat{I}|} \sum_{\substack{z_i \in \mathcal{Z}_2 \\ z_i \sim \mathbb{P}^\star}} z_i \mathbb{I}\left\{z_i \in \widehat{I}\right\} - \mu\right|}_{T_1} + \underbrace{\left|\frac{1}{|\widehat{I}|} \sum_{\substack{z_i \in \mathcal{Z}_2 \\ z_i \sim Q}} z_i \mathbb{I}\left\{z_i \in \widehat{I}\right\} - \mu\right|}_{T_2}.$$

**Bounding $T_1$:** From [41, Claim 6], we bound $T_1$ with probability at least $1 - \delta_3 - \delta_5$ as

$$T_1 \leq \frac{\epsilon + f_n(\epsilon, \delta_5)}{1 - \delta_4} \cdot 4\sqrt{\sigma^2 \log\left(\frac{1}{\delta_1}\right)},$$

where $\delta_4 = (\delta_1 + \epsilon) + f_n(\delta_1 + \epsilon, \delta_3)$.

**Bounding $T_2$:** To bound $T_2$, we split the terms further.

$$T_2 = \left| \frac{1}{|\widehat{I}|} \sum_{\substack{z_i \in \mathcal{Z}_2 \\ z_i \in \widehat{I} \\ z_i \sim Q}} (z_i - \mu) \right| = \frac{|\widehat{I}_{\mathbb{P}^\star}|}{|\widehat{I}|} \left| \frac{1}{|\widehat{I}_{\mathbb{P}^\star}|} \sum_{\substack{z_i \in \mathcal{Z}_2 \\ z_i \in \widehat{I} \\ z_i \sim Q}} (z_i - \mu) \right|$$

$$\leq \underbrace{\frac{|\widehat{I}_{\mathbb{P}^\star}|}{|\widehat{I}|} \left| \left( \frac{1}{|\widehat{I}_{\mathbb{P}^\star}|} \sum_{\substack{z_i \in \mathcal{Z}_2 \\ z_i \in \widehat{I} \\ z_i \sim Q}} z_i \right) - \mathbb{E}[x | x \in \widehat{I}, x \sim \mathbb{P}^\star] \right|}_{T_{2,1}}$$

$$+ \underbrace{\frac{|\widehat{I}_{\mathbb{P}^\star}|}{|\widehat{I}|} \left| \mathbb{E}[x | x \in \widehat{I}, x \sim \mathbb{P}^\star] - \mu \right|}_{T_{2,2}},$$

where $|\widehat{I}_{\mathbb{P}^\star}| = \sum_{\substack{z_i \in \mathcal{Z}_2 \\ z_i \sim \mathbb{P}^\star}} \mathbb{I}\left\{ z_i \in \widehat{I} \right\}$ is the number of elements in $\mathcal{Z}_2$ that were originally sampled from $\mathbb{P}^\star$.

$T_{2,1}$ is the deviation of the mean of the samples originally sampled from $Q$ and remain in $\widehat{I}$ from the mean of $\mathbb{P}^\star$ conditioned on the event that they belong to $\widehat{I}$ as well. $T_{2,2}$ measures the deviation of the mean of $\mathbb{P}^\star$ from the mean of the same distribution conditioned on $\widehat{I}$.

**Bounding $T_{2,1}$:** We bound $T_{2,1}$ using [41, Lemma 15]. With this result, we get that with probability at least $1 - \delta_7$,

$$T_{2,1} \leq \sqrt{\frac{2\sigma^2 \log(3/\delta_7)}{\mathbb{P}^\star(\widehat{I})}} + 2\sqrt{\sigma^2 \log\left(\frac{1}{\delta_1}\right) \frac{\log(3/\delta_7)}{|\widehat{I}_{\mathbb{P}^\star}|}}.$$

**Bounding $T_{2,2}$:** To control $T_{2,2}$ we make use of Proposition 8 in conjuction with [41, Lemma 14] to get

$$T_{2,2} \leq 2\mathbb{P}^\star(\widehat{I}^c) \sqrt{\sigma^2 \log\left(\frac{1}{\mathbb{P}^\star(\widehat{I}^c)}\right)},$$

where $\mathbb{P}^\star(A)$ is the probability that $z \sim \mathbb{P}^\star$ lies in $A$. Finally, we bound $\mathbb{P}^\star(\widehat{I}^c$ using [41, Claim 7] to obtain with probability at least $1 - \delta_6$ that

$$\mathbb{P}^\star(\widehat{I}^c) \leq C_1 \epsilon + C_2 \delta_1 + C_3 \frac{\log(n)}{n} + C_4 \frac{\log(1/\delta_6)}{n} + C_5 \frac{\log(1/\delta_3)}{n},$$

where $\{C_i\}_{i=1}^6$ are universal constants.

Therefore, combining the bounds for $T_1$, $T_{2,1}$ and $T_{2,2}$, and setting $\delta_1 = \epsilon$, $\delta_3 = \delta_5 = \delta_6 = \delta_7 = \delta/4$ and noting that for the choice of $n$ $|\widehat{I}_{\mathbb{P}^\star}| \geq \frac{n}{2}$, we get the final deviation bound:

$$T_1 + T_{2,1} + T_{2,2} \lesssim \epsilon \sqrt{\sigma^2 \log\left(\frac{1}{\epsilon}\right)} + \sqrt{\sigma^2 \log\left(\frac{1/\delta}{n}\right)},$$

and this completes the proof of the lemma. $\qquad\square$

# F Proof of Theorem 3

In this section, we present the proof of Theorem F. The proof mostly follows the analysis in the proofs of Lemma 7 and Theorem 1. The only difference is that we will not be able to use the sub-Gaussianity of Ising model distributions anymore, as it is no longer applicable.

*Proof.* Following the proof of Lemma 7, we have for any $v$ such that $\|v\|_1 = 1$ that

$$
\left| \left\langle v, \nabla \mathcal{L}_{\theta^{(1)},i}(2\theta^{(2)}(i)) \right\rangle \right| \leq \left| \mathbb{E}_{z_1,z_2 \sim \mathcal{C}} \left[ \left\langle v, \nabla \ell_i(2\theta^{(2)}(i); z_1) \right\rangle \mathbb{I}\{z_1 \neq z_2\} \right] \right|
$$
$$
+ \left| \mathbb{E}_{z_1,z_2 \sim \mathcal{C}} \left[ \left\langle v, \nabla \ell_i(2\theta^{(2)}(i); z_2) \right\rangle \mathbb{I}\{z_1 \neq z_2\} \right] \right|
$$
$$
\overset{(i)}{\leq} \underbrace{\mathbb{E}_{z_1,z_2 \sim \mathcal{C}} \left[ \left| \left\langle v, \nabla \ell_i(2\theta^{(2)}(i); z_1) \right\rangle \right| \mathbb{I}\{z_1 \neq z_2\} \right]}_{T_1}
$$
$$
+ \underbrace{\mathbb{E}_{z_1,z_2 \sim \mathcal{C}} \left[ \left| \left\langle v, \nabla \ell_i(2\theta^{(2)}(i); z_2) \right\rangle \right| \mathbb{I}\{z_1 \neq z_2\} \right]}_{T_2},
$$

where in step $(i)$, we have used Jensen's inequality for $f(x) = |x|$.

**Bounding $T_1$:** By Hölder's inequality $\left| \left\langle v, \nabla \mathcal{L}_{\theta^{(1)},i}(2\theta^{(2)}(i)) \right\rangle \right| \leq \|v\|_1 \left\| \nabla \mathcal{L}_{\theta^{(1)},i}(2\theta^{(2)}(i)) \right\|_\infty$. Again by Jensen's inequality, and the explicit form of $\nabla \ell_i$, we have $\left\| \nabla \mathcal{L}_{\theta^{(1)},i}(2\theta^{(2)}(i)) \right\|_\infty = \left\| \mathbb{E}\left[ \nabla \ell_i(2\theta^{(2)}(i), z_1) \right] \right\|_\infty \leq \mathbb{E}\left[ \|\nabla \ell_i(2\theta^{(2)}(i), z_1)\|_\infty \right] \leq 1$. Therefore,

$$
\mathbb{E}_{z_1,z_2 \sim \mathcal{C}} \left[ \left| \left\langle v, \nabla \ell_i(2\theta^{(2)}(i); z_1) \right\rangle \right| \mathbb{I}\{z_1 \neq z_2\} \right] \leq \mathbb{E}_{z_1,z_2 \sim \mathcal{C}} \left[ \mathbb{I}\{z_1 \neq z_2\} \right] \leq \epsilon.
$$

**Bounding $T_2$:** $T_2$ can be bounded in the exact same way as $T_1$.

Plugging these bounds, we get that

$$
\left| \left\langle v, \nabla \mathcal{L}_{\theta^{(1)},i}(2\theta^{(2)}(i)) \right\rangle \right| \leq 2\epsilon.
$$

Now, following the first part of the proof of Theorem 1, we have using Hölder's inequality and the bound above that

$$
\frac{C}{2} \frac{\exp(-O(\omega))}{\omega} \|\Delta_i\|_2^2 \leq \left| \left\langle \Delta_i, \nabla \mathcal{L}_{\theta^{(1)},i}(2\theta^{(2)}(i)) \right\rangle \right| \leq \|\Delta_i\|_1 \left\| \nabla \mathcal{L}_{\theta^{(1)},i}(2\theta^{(2)}(i)) \right\|_\infty \leq 2\epsilon \|\Delta_i\|_1
$$

where $\Delta_i = 2\theta^{(1)}(i) - 2\theta^{(2)}(i)$. Now, since $\theta^{(1)}$ and $\theta^{(2)}$ are parameters of Ising models with maximum vertex degree $d$, $\Delta_i = 2\theta^{(1)}(i) - 2\theta^{(2)}(i)$ has atmost $2d$ non-zero elements. Consequently, we get $\|\Delta_i\|_1 \leq \sqrt{d}\|\Delta_i\|_2$.

Finally, plugging the above norm inequality in the previous bound, we have:

$$
\|\Delta_i\|_2 \lesssim \epsilon\sqrt{d}\omega \exp(O(\omega)).
$$

Analogously, since $d \leq k$ when $G^{(1)}, G^{(2)} \in \mathcal{G}_{p,k}$, we have

$$
\|\Delta_i\|_2 \lesssim \epsilon\sqrt{k}\omega \exp(O(\omega)),
$$

Alternatively, note that by the triangle inequality: $\|\Delta_i\|_1 \leq \|2\theta^{(1)}(i)\|_1 + \|2\theta^{(2)}(i)\|_1 \leq 4\omega$. This gives us:

$$
\|\Delta_i\|_2 \lesssim \sqrt{\epsilon}\omega \exp(O(\omega))
$$

Since both types of inequalities holds simultaneously, we recover the statements of the theorem for $\mathcal{G}_{p,d}$ and $\mathcal{G}_{p,k}$. □