[Reviews · NeurIPS 2020]

Review 1

Summary and Contributions: The paper presents sample complexity results for estimating the parameters of Ising models under Huber's contamination model. In particular, it is assumed that samples are drawn from a distribution: (1-eps) P_theta + eps Q where P_theta is an Ising model without external fields and Q is an arbitrary distribution over {-1, 1}^p. As usual, the Ising model density P_theta(x) over the hypercube is proportional to exp( x^T theta x). The Ising model P_theta to be learned is assumed to be in high temperature, satisfying Dobrushin's condition. The goal is to estimate, to within some small ell_2 error, the vector of theta_ij parameters at each node i of the Ising model, i.e. for every node i, estimate theta_i=(theta_ij)_j The paper provides methods to do so whose error scales as robustness error + sampling error The first term is eps \sqrt{log(1/eps)} while the second takes the form - sqrt{ k log p /n}, if the Ising model has k edges or - sqrt{d log p/n}, when every node has degree d So the paper provides sparse regression type bounds for the theta_i's from a contaminated distribution. Unfortunately, none of these methods are computationally efficient. In terms of techniques the most interesting contribution in my view is probably Theorem 1 (in Section 2/appendix B) which translates from a TV bound between two Ising models to an ell_2 bound for their corresponding theta_i vectors, under Dobrushin's condition. The translation from a TV bound to a parameter bound analyzes the conditional log likelihood of the sample at a node conditioning on the other nodes. That function is concave and is maximized at the true parameters, and the argument proceeds by analyzing its strong concavity and bounding its gradient. The proof uses recent results proven about the Ising model under Dobrushin's condition, and this approach of comparing the gradient and the Hessian has also been done in recent papers on Ising models. In any event, it is a good bound to know. Another interesting insight (in Section 4/appendix D) is combining the afore-mentioned recent results about the Ising model, the strong concavity of the conditional log-likelihood and robust mean estimation in order to do robust l1 regression at every node.

Strengths: Learning under sample contamination is an interesting problem for ML and Statistics. Ising models are a prominent family of distributions and Huber's contamination model is a standard model. The paper provides sparse regression error bounds under Huber's model which are welcome. It also provides a structural result translating TV error to parameter error, and an approach to robust sparse l1 regression around every node.

Weaknesses: The algorithms are inefficient, and the techniques are not very novel in my view: -Analyzing the log likelihood around the optimum by bounding its strong convexity and its gradient isn't too novel an approach - Using the Yatracos class to learn Ising models has been employed in a recent paper: https://arxiv.org/abs/1806.06887 - In fact, all the technical staff in Section 3 can be circumvented as follows: 1. It is easy to construct an epsilon net in TV distance of all Ising models with at most k edges of size at most (p^2 choose k) (k/eps^2)^k (the way to do this is to use the exact expression for Symmetric KL distance between two Ising models, Pinsker's inequality, and gridding of all edge parameters) 2. Whenever a eps-net of size N exists for a family of distribution, it is known that logN / eps^2 samples from a target distribution in the family suffice to select a distribution from the net that is eps close to the target the error rate of sqrt{k log(p)/n} for learning in TV distance follow from the above observations.. In any event, obtaining the TV to parameter bound and doing robust sparse regression at each node hasn't been done before as far as I know, and it employs some heavy machinery recently shown for the Ising model.

Correctness: Yes

Clarity: Yes

Relation to Prior Work: Yes

Reproducibility: Yes

Additional Feedback:


Review 2

Summary and Contributions: The paper proposed a method for learning an Ising model from data samples from a contamined model (a mixture of an Ising model and an arbitrary distribution).

Strengths: The contamination model of data samples is well-known and reasonable (Huber).

Weaknesses: The condition \max_{u \in V} \sum_{v \in V} |\theta_{uv}| < 1 from Line 78 and 89 seems too restrictive. The proposed estimators do not recover the true model as n goes to infinity. The proposed estimators are not computationally efficient. (Please see my additional feedback below.)

Correctness: The main claims seem correct.

Clarity: The paper is clear.

Relation to Prior Work: Relation to prior work is properly discussed.

Reproducibility: Yes

Additional Feedback: Please discuss the condition \max_{u \in V} \sum_{v \in V} |\theta_{uv}| < 1 from Line 78 and 89. For a growing number of nodes in V, this condition seems very restrictive because of the summation across all nodes. The proposed estimators do not recover the current model as n goes to infinity. In Lemma 2 and Corollary 2, there are terms 2\epsilon and 2\epsilon\sqrt{log(1/\epsilon)} respectively, which are constant with respect to n. Similarly, more complicated terms are in Theorem 2 and Corollary 3. Is this related to Lemma 1? Please clarify. Estimators in Eq.(8) and Eq.(10) are not computationally efficient. Can the authors provide more insight into #P-hardness? I can see that in both cases the \argmin and the \sup are hard to efficiently compute. === AFTER REBUTTAL I am satisfied with the response regarding the Dobrushin condition (\max_{u \in V} \sum_{v \in V} |\theta_{uv}| < 1). I am also satisfied with the response to the estimator inconsistency, specifically the clarification regarding Lemma 1. The authors state that their estimations are "exponential time" (i.e., worse than #P-hard). Still I believe the results can be interesting for the community. Given the above, I raise my evaluation from 6 to 7.


Review 3

Summary and Contributions: The paper studies the problem of robustly learning discrete graphical models where a small fraction of the samples can be arbitrarily corrupted (known as Huber’s contamination model). It focuses on Ising models where prior work provided partial results. The authors operate (for most of the paper) under the Dobrushin’s uniqueness regime where they are able to obtain a bound on the l2 distance between the underlying parameters vectors of two Ising models which are epsilon-close in Total Variation (TV) distance. They use this together with the observation that under epsilon Huber’s contamination, the TV between corrupted and true distribution is at most epsilon; to show that in the asymptotic setting a minimum distance estimator can recover the parameter vector and edge structure (when minimum edge strength is large enough) for graphs on p nodes with (i) bounded (k) number of edges, (ii) bounded degree (d). Then they move to showing how an approximation of the minimum distance estimator (inspired by work of Yatracos) works in the non-asymptotic setting for graphs with bounded number of edges. For graphs with bounded degree in the non-asymptotic setting: they propose an estimator which performs a robust sparse logistic regression to recover the parameters of the uncontaminated model.

Strengths: The paper studies a problem of high relevance to NeurIPS community and provides a tight understanding of the information theoretic complexity of the problem for specific graph families. The claims are theoretically sound. As a by-product the paper also provides a modulus of continuity bound for Ising models under Dobrushin’s condition which is the first result of its kind for this regime. This is a non-trivial and novel statement which has other potential implications beyond the problem focused on in this work.

Weaknesses: A big part of the challenge of robust estimation is computationally efficient algorithms and unfortunately the paper does not address the computational complexity of the task. The proposed algorithms in the paper seem to require an exponential amount of time to perform. Given the potential computational inefficiency of the algorithms, the improvement over previously known bounds in the non-contaminated case is less impressive as the previously known bounds cited in the paper are shown for computationally efficient algorithms.

Correctness: I have not verified all the claims but they appear correct.

Clarity: Yes

Relation to Prior Work: Yes

Reproducibility: Yes

Additional Feedback: Questions/Suggestions: 1. Could you comment on the computational complexity of your estimators for G_p,d and G_p,k in the non-asymptotic setting? 2. In the non-contaminated setting, it is known that model width works as a more relaxed parameter of complexity of learning as opposed to maximum degree. Can something similar be said here? 3. It seems like one of the significant contributions of the paper is the modulus of continuity bound under Dobrushin’s condition. Where else within the Ising model literature might this bound have applications? Minor Typos: 1. Line 208: “leadins” -> “leads” ------ Post-Author Feedback------ Thank you to the authors for providing answers to some of my questions. I have carefully considered the response and I do feel the lack of computational efficiency is still a weakness although a statistical understanding of what is possible is an important first step. Given that, the modulus of continuity bound they provide has other applications which they expand upon in their response. This is an interesting result the authors show which makes a case for the acceptance of the paper. Overall, I am inclined to maintain my score for the paper.


Review 4

Summary and Contributions: This paper considers the task of information-theoretically learning Ising models with an unknown graph structure in Huber’s eps-contamination model. The paper characterizes the TV modulus of continuity for learning Ising models in Theorem 1 and Lemma 1, which yield information theoretic lower bounds on the possible rates of estimation in terms of eps. It then provides and analyzes two different inefficient algorithms for estimation that achieve the optimal estimation rate in eps: (1) a TV projection estimator and (2) an algorithm estimating the neighborhood of each node with robust sparse logistic regression. These estimators are analyzed in the two different parameter regimes where the total number of edges of the underlying graph is bounded by a parameter k and when the max degree is bounded by d. As discussed in the paper, the estimator also recovers an improved information-theoretic upper bound in latter case even when eps = 0, in particular obtaining a better dependence on d. UPDATE: Thank you to the authors for the response and changes. While Dobrushin's condition is popular, it's unclear that it's necessary in this information-theoretic rate a priori (maybe especially given that the Lindgren result does not need this?). However, it is satisfying that the paper has some partial results towards removing Dobrushin's condition in the appendix. My review remains that this paper should be accepted.

Strengths: The paper involves a number of interesting techniques and proofs. The proof of Theorem 1 seems nontrivial and is a large part of the supplementary material. As mentioned above, the information-theoretic upper bounds in this paper seem to improve upon the best known upper bounds even in the uncontaminated case. It seems as though the only known upper bounds in the contaminated case are the computationally efficient upper bounds in Lindgren et al. that achieve an error of sqrt(eps). So, this paper improves upon the best known information theoretic upper bounds substantially.

Weaknesses: The paper restricts its attention to the high-temperature regime. A priori, recent results for learning Ising models do not seem to need this kind of assumption. In contrast, the paper does not seem to justify that assuming Dobrushin’s conditions and being in the high-temperature regime is necessary. In many robust problems, Huber eps-contamination often yields an information-theoretic lower bound of \Omega(eps) on the rate of estimation of the model parameters. For example, this is the case in robust mean estimation, robust linear regression, robust PCA and other similar problems. The fact that this is also the case for Ising models is not surprising. Proofs of these facts in other problems have also established TV moduli of continuity. However, the proof of Theorem 1 here seems to be technically more challenging than analogous proofs for other problems in the literature.

Correctness: The proofs appear to be correct and are well-explained. While this is a purely theoretical paper, it does have synthetic experiments in Section 5 which seem correct and help the reader understand the paper’s theoretical results.

Clarity: The paper is very well written and organized. This is a strength of the paper.

Relation to Prior Work: The relation of the paper to prior work is well-explained in Section 1.1. A more detailed comparison of the upper and lower bounds in Lindgren et al. to those in this work would be helpful. It may also be helpful to discuss in more detail known information-theoretic lower bounds on the sample complexity in terms of k, d, lambda and omega, even if only in the uncontaminated setting for the task of learning uncorrupted Ising models.

Reproducibility: Yes

Additional Feedback: Pg. 3 Line 105, the TV equality should be an inequality

[Author Response · NeurIPS 2020]

We thank the reviewers for their kind comments, and for their consensus view that our theoretical results on TV modulus of continuity (Theorem 1) and statistically optimal robust learning of degree bounded Ising Models (Theorem 4) are novel, non-trivial and interesting. We are also thankful for the reviewers' concrete suggestions on improving the draft, which we will incorporate in the final version of our work. We begin by addressing two common concerns raised by reviewers before addressing specific questions raised by each reviewer.

**Computational Efficiency.** We agree with the reviewers that our proposed estimators are not computationally efficient. In particular, our estimator proposed in Equation 10 runs in exponential time due to the sizes of the coverings that the optimization problem is defined over. However, note that in stark contrast to many other problems studied in robust statistics, even ignoring considerations of computational efficiency, basic statistical questions of the fundamental limits of robust estimation for Ising models are not well-understood. Our main contribution is in giving statistically optimal estimators, thereby, establishing tight information theoretic rates for estimation of Ising models under contamination. Designing polynomial-time estimators for robust learning of Ising models that are also statistically optimal is certainly an interesting open problem, and we leave that for future work.

**Dobrushin's condition.** We work in the high-temperature regime i.e., $\max_{u \in V} \sum_{v \in V} |\theta_{uv}| < 1$. Note that while this may seem restrictive, this assumption is widely popular for studying Ising Models, for example, see related works in statistical physics [2, 5], mixing times of Glauber Dynamics [3, 1], correlation decay [4], and more recently in estimation and testing problems [Daskalakis et al. (2019), Dagan et al. (2020)]. Note that in Appendix A, we also provide preliminary results for Ising models which don't satisfy the Dobrushin condition.

**AR1:** *Related Work and TV-Cover.* We thank the reviewer for pointing us to a recent work by Devroye et al, and in particular, for providing an alternate exponential time estimator for $G_{p,k}$ based on TV-covers. We will update our manuscript add the relevant citation and also add discussion of this alternate exponential time estimator.

**AR2:** *Inconsistent Estimator.* We agree with the reviewer that our estimator doesn't recover the true model even in the infinite sample limit. However, note that this is not a limitation of our work, and is true in general (even for the most basic problem of mean estimation) in Huber's model. In particular, as shown in our lower bound (Lemma 1) , there exists two Ising models whose TV distance is $O(\epsilon)$, but the parameters are $\Omega(\epsilon)$ far apart. This shows that in the contaminated setting, one cannot hope to recover the exact model parameters. The interesting question then becomes whether we could recover the parameters up to this unavoidable bias due to Huber contamination: which we show our estimators are able to do.

**AR3:** *Model Width as relaxed parameter.* We thank the reviewer for raising this subtle issue. Note that under the high-temperature regime, the model width is upper bounded by 1, thereby, making it a constant. However, in Appendix A, we present additional preliminary results for Ising models with unrestricted model width $\omega$, and seek optimal dependence on $\omega$ as well.

*Applications of Modulus of Continuity.* We believe that our modulus of continuity analysis can improve several existing results. For example, in the uncontaminated setting, our result directly improves the sample complexity analysis of the maximum likelihood graph decoder in Santhanam and Wainwright (2012). Moreover, we believe that our result can also improve testing of Ising Models, either by improving the sample complexity of existing symmetric KL-divergence based algorithm of Daskalakis et al (2018), or by using the contrapositive of our Theorem, to potentially design new tests based on $\ell_2$ error of neighborhood vectors.

**AR4:** *Comparison to Lindgren et al.* Lindgren et.al. work in a more general setting, but do not distinguish between high and low-temperature regimes. In particular, they provide an estimator which achieves an upper bound of $O(\sqrt{\epsilon})$ and a lower bound of $\Omega(\epsilon)$. In contrast, our results show improved error rates in the high-temperature regime, and match their results in the low-temperature one (See Appendix A).

We thank the reviewer for their suggestions and will certainly include a more detailed exposition of known information theoretic limits of learning Ising models in the uncontaminated setting, highlighting the dependencies on $\lambda, \omega, k/d$.

**Additional references:**

**(1)** De Sa, Olukotun and Ré. Ensuring rapid mixing and low bias for asynchronous Gibbs sampling, 2016.

**(2)** Dobrushin and Shlosman. Completely analytical interactions: constructive description, 1987.

**(3)** Kulske. Concentration inequalities for functions of Gibbs fields with application to diffraction and random Gibbs measures, 2003.

**(4)** Kunsch. Decay of correlations under Dobrushin's uniqueness condition and its applications, 1982.

**(5)** Stroock and Zegarlinski. The logarithmic Sobolev inequality for discrete spin systems on a lattice, 1992.


[Meta-Review · NeurIPS 2020]

All the reviewers are positive about the paper. Especially after the rebuttal that addressed a number of concerns. I strongly suggest authors to incorporate received feedback into the final version of the manuscript.